# Probiotics: Protecting Our Health from the Gut

**DOI:** 10.3390/microorganisms10071428

**Published:** 2022-07-14

**Authors:** Gael Urait Varela-Trinidad, Carolina Domínguez-Díaz, Karla Solórzano-Castanedo, Liliana Íñiguez-Gutiérrez, Teresita de Jesús Hernández-Flores, Mary Fafutis-Morris

**Affiliations:** 1Doctorado en Ciencias Biomédicas, Con Orientaciones en Inmunología y Neurociencias, Universidad de Guadalajara, Sierra Mojada 950, Guadalajara 44340, Mexico; gael.varela@alumnos.udg.mx (G.U.V.-T.); carolina.dominguezd@alumnos.udg.mx (C.D.-D.); 2Centro de Investigación en Inmunología y Dermatología (CIINDE), Calzada del Federalismo Nte 3102, Zapopan 45190, Mexico; 3Doctorado en Ciencias de la Nutrición Traslacional, Universidad de Guadalajara, Sierra Mojada 950, Guadalajara 44340, Mexico; karla.solorzano1972@alumnos.udg.mx; 4Instituto de Investigación de Inmunodeficiencias y VIH, Hospital Civil de Guadalajara, Coronel Calderón 777, Guadalajara 44280, Mexico; liliana.iniguez@alumnos.udg.mx (L.Í.-G.); teresita.hflores@academicos.udg.mx (T.d.J.H.-F.); 5Departamento de Disciplinas Filosóficas Metodológicas e Intrumentales, Centro Universitario de Ciencias de la Salud, Universidad de Guadalajara, Sierra Mojada 950, Guadalajara 44340, Mexico; 6Departamento de Fisiología, Centro Universitario de Ciencias de la Salud, Universidad de Guadalajara, Sierra Mojada 950, Guadalajara 44340, Mexico

**Keywords:** intestinal microbiota, dysbiosis, microbiota–gut–brain axis, microbiota–gut–skin axis, microbiota–gut–lung axis, microbiota–gut–heart axis, microbiota–metabolism, probiotics and postbiotics

## Abstract

The gut microbiota (GM) comprises billions of microorganisms in the human gastrointestinal tract. This microbial community exerts numerous physiological functions. Prominent among these functions is the effect on host immunity through the uptake of nutrients that strengthen intestinal cells and cells involved in the immune response. The physiological functions of the GM are not limited to the gut, but bidirectional interactions between the gut microbiota and various extraintestinal organs have been identified. These interactions have been termed interorganic axes by several authors, among which the gut–brain, gut–skin, gut–lung, gut–heart, and gut–metabolism axes stand out. It has been shown that an organism is healthy or in homeostasis when the GM is in balance. However, altered GM or dysbiosis represents a critical factor in the pathogenesis of many local and systemic diseases. Therefore, probiotics intervene in this context, which, according to various published studies, allows balance to be maintained in the GM, leading to an individual’s good health.

## 1. Intestinal Microbiota and Dysbiosis

The human body is in direct contact with approximately 3.9 × 10^13^ bacteria, which colonize all surfaces and cavities that come into contact with the external environment and with which we must maintain a symbiotic relationship for the proper functioning and health of our bodies. For this reason, the term human microbiota (HM) is used to refer to these microorganisms, including bacteria, fungi, protozoa, and viruses, among others [1]. Since the proportion of bacteria is two or three times greater than that of any other microbe, studies have focused mainly on the analysis of the composition and functions of bacterial communities [2,3]. It is estimated that the ratio of bacteria to human cells is 1:1 and that bacteria are responsible for approximately 3.3 million nonself genes, which implies a contribution of 150 times more genetic information than the human genome itself [3,4,5]. This has led to the HM being considered the “hidden organ” of the human body. Additionally, the HM participates in various physiological functions and metabolic activities that benefit the host [5].

The HM comprises several distinct microbial communities found on epithelial and mucosal surfaces that are in contact with the outside environment, such as skin, the oral cavity, conjunctiva, and the respiratory, genitourinary, and gastrointestinal (GI) tracts [2]. Since the GI tract has one of the largest epithelial surfaces in the human body, with a surface of 250 to 400 m^2^ [6], it is estimated that 70% of all bacteria in the human body are found in the colon alone [1], which are referred to as the gut microbiota (GM) [2,7]. There are more than 500 species of bacteria in the intestine, the most common belonging to the phyla Bacteroidetes and Firmicutes, followed by Proteobacteria, Actinobacteria, and Verrucomicrobiota [8,9].

The metabolic functions of the GM transform dietary substrates and generate new metabolites, polysaccharides, short-chain fatty acids (SCFAs), vitamin K, folic acid, and amino acids such as arginine and glutamine [4,10]. Furthermore, the GM has an important immunomodulatory role since it affects the immune system by providing physical protection and preventing the entry of bacteria, antigens, proinflammatory factors, or other metabolites. It also promotes tolerance to bacteria through the induction of regulatory T cells (Treg) and anti-inflammatory cytokines to maintain a balanced immune response [6].

Symbiotic interactions between GM and the host contribute to maintaining intestinal homeostasis and influence the innate and adaptive immune response, which gives these interactions a key role in the regulation of our health.

### 1.1. Intestinal Dysbiosis and Its Health Consequences

Factors such as diet, age, antibiotic consumption, tobacco consumption, and lifestyle, among others, can impact and modify the composition and function of the GM [11,12]. This alteration in the GM is termed intestinal dysbiosis (ID) and is typically characterized by the proliferation of pathogens and a loss of diversity and commensal bacteria [12].

Pathogens, which are bacteria that can harm the host through virulence factors, and pathobionts, which are commensal bacteria that can become harmful in certain circumstances, increase their abundance during ID and disrupt homeostatic and metabolic processes in the host [13,14]. In a dysbiotic state, the pathogenic bacteria produce proinflammatory metabolites, harmful secondary bile acids such as deoxycholic and lithocholic acid, and hydrogen sulfide, all of which contribute to the exacerbation of inflammatory conditions in the gut epithelium [15].

In particular, ID plays a role not only in the pathologies of the gastrointestinal tract but also in diseases that originate in the extraintestinal organs [12]. The altered GM can communicate and interact with the immune system through these harmful metabolites, thereby increasing the number of proinflammatory cells, cytokines, and metabolites that enter the bloodstream and arrive in distal organs such as the brain, lungs, heart, and skin, where they contribute to the inflammatory state of these organs.

The interactions between the GM and extraintestinal organs have been grouped and are referred to as interorganic axes [16]. It is now well established that the GM plays an essential role in the development or alleviation of diseases in these specific organs outside the intestinal tract by providing an alternative means of worsening inflammation or improving symptoms.

### 1.2. Prebiotics, Probiotics, Synbiotics, and Postbiotics

It has been well established that restoring the balance of the GM favors dynamism and the functioning of the immune system [17]. In recent years, biotics have been researched due to their ability to modulate the GM in order to improve the health status of the host. The term biotics refers to prebiotics, probiotics, synbiotics, and postbiotics, which have an impact on homeostasis.

Prebiotics are the food for the bacteria in the microbiota. The best known are inulin and fructooligosaccharides, which stimulate the proliferation of *Bifidobacterium* spp. and *Lactobacillus* spp. [18]. On the other hand, the term probiotics refers to bacteria that exert beneficial functions on the microbiota. They are live microorganisms that have health benefits for the host in great enough quantities [19]. Strains of *Bifidobacterium* and *Lactobacillus* can inhibit pathogen colonization, improve barrier function, and modulate the immune response [20,21]. Synbiotics refer to the functional combination of probiotics and prebiotics, which, through their interaction, have a positive effect on the host [18].

Finally, postbiotics are specific bioactive compounds released by microorganisms through their metabolic activity that also exert a beneficial effect on the host [22]. This group includes, but is not limited to, organic acids, carbohydrates, lipids, proteins, vitamins, surface proteins, and complex molecules such as lipoic acids and derivatives of peptidoglycans [23].

In this review, we analyze the most studied axes that involve the GM and extraintestinal organs, such as the microbiota–gut–brain axis, microbiota–gut–skin axis, microbiota–gut–lung axis, microbiota–gut–heart axis, and the interaction of the microbiota with metabolic disease, such as obesity and diabetes mellitus. We also discuss how the pathologies of these organs are affected during ID. Finally, we consider previous research that has administered oral probiotics as an alternative or complementary treatment and whether their use improves the symptoms of diseases related to these axes.

## 2. Microbiota–Gut–Brain Axis (MGB)

The nervous system consists of the central nervous system (CNS), including the brain and spinal cord, and the peripheral nervous system (PNS), which consists of the ganglia that give rise to nerve branches connected to the organs of the human body [24]. The PNS is divided into the somatic nervous system and the autonomous nervous system. The latter is considered involuntary and has several variants, including the enteric nervous system (ENS), which integrates various sympathetic and parasympathetic nerve fibers that enable bidirectional communication between the CNS and the GI tract. The practical cooperation between the two systems has led to the ENS being considered the second brain, or the little brain, of the human body [25,26].

The ENS extends throughout the digestive tract and is divided into two plexuses, the submucosal and the myenteric. These plexuses are organized by different types of neurons that control motility and peristalsis and even regulate the immunity of the intestinal mucosa [27]. The millions of bacteria that make up the GM are close to the ENS. It has even been observed that a communication system exists between gut bacteria and the CNS. These findings have led to the description of the “microbiota–gut–brain axis.” This axis includes connections between different biological systems to regulate homeostatic processes at the GI level, in the CNS, and in the gut bacteria themselves [16,28].

The microbiota–gut–brain axis has successfully linked the contents of the GM to various neurodegenerative disorders, developmental disorders, and mood changes [29,30,31,32,33,34,35,36]. The relationship between these diseases and the microbiota is mainly related to several bacterial metabolites that act as immunoregulatory and neurochemical factors [37,38,39].

The bacterial metabolites involved in regulation mainly include SCFAs, various tryptophan metabolites, and microbial neurotransmitters (Figure 1) [37]. SCFAs consist of fatty acids with no more than six carbon atoms, with the most significant proportions being acetate (C2), propionate (C3), and butyrate (C4). SCFAs have the ability to cross the blood–brain barrier (BBB) or bind to the receptors located there. In addition, SCFAs promote the secretion of hormones from intestinal endocrine cells; due to these properties, they may regulate various functions at the CNS level and may even be involved in the development of neurological disorders [37,40]. Tryptophan metabolites, for their part, have been described as molecules with a high regulatory potential that send signals from the GM to various peripheral organs, including the brain, where they can regulate the development and activation of microglia and induce the production of molecules to modulate inflammation in the CNS [41,42]. Finally, microbial neurotransmitters include molecules such as γ-aminobutyric acid (GABA), a molecule that can modulate brain function and inflammatory processes associated with the CNS, and catecholamines, which can regulate cognitive abilities, mood, and gut motility [27,43].

Given the link between alterations in the GM and neurological disorders, the use of probiotic microorganisms capable of producing the microbial metabolites associated with the regulation of various CNS disorders has emerged as a therapeutic strategy. In the following sections, we will address the relationship between certain diseases and the processes of gut dysbiosis and how the use of probiotics has been shown to be a potential therapeutic tool for these diseases.

### 2.1. Alzheimer’s Disease

Alzheimer’s disease (AD) is one of the most common neurodegenerative diseases. It is a progressive condition that impairs cognitive function [44]. AD is based on the accumulation of amyloid plaques that promote a state of chronic inflammation. These plaques are characterized by the increased expression of a misfolded β-amyloid protein (Aβ), tau proteins, and intraneuronal neurofibrillary tangles, which promote the loss and atrophy of neuronal development in addition to neuroinflammation [45,46].

Dysbiosis is an essential factor in the development or exacerbation of AD. Several studies have linked alterations in the composition of the GM and the presence of pathogenic microorganisms in the gut to AD development [5,34,47,48].

Some studies on the relationship between dysbiosis and AD have shown, by using sequencing techniques, that the GM composition differs between AD patients and control subjects matched for age and sex. Among the differences, a decreased abundance of Firmicutes and *Bifidobacterium* and an increase in the abundance of Bacteroidetes bacteria were found in AD patients [49]. Other reports have described an increase in the fecal serum concentration of calprotectin, a protein associated with gut inflammation, resulting in impaired barrier function in AD patients compared with control subjects [50].

In animal models, changes in GM induced by endotoxin administration in C57BL/6J mice have been shown to promote Aβ production in the brain and also cause BBB dysregulation and cognitive decline in mice [51,52]. Furthermore, gut dysbiosis in a *Drosophila* model of AD resulted in the exacerbation of the disease, leading to neuronal loss, increased apoptosis, decreased locomotor activity, and decreased life expectancy [53].

Supplementation with probiotics and their metabolites effectively restores the altered GM and ameliorates AD symptoms. Among the probiotics studied, the administration of *Lactiplantibacillus*
*plantarum* DP189 prevented cognitive dysfunction and increased the levels of microbial neurotransmitters, such as serotonin, dopamine, and GABA; this allowed an improvement in neuronal damage as well as a reduction in Aβ deposition [54]. Similarly, the administration of a probiotic mix containing *Lactobacillus acidophilus*, *Lacticaseibacillus casei* (basonym of *Lactobacillus casei*), *Bifidobacterium bifidum*, and *Limosilactobacillus fermentum* (basonym of *Lactobacillus fermentum*) was successful in improving cognitive function as well as metabolic state in AD patients [55]. Other probiotics have also shown the ability to restore the GM composition, improve cognitive impairment, and reduce Aβ deposition. These probiotics include *B**ifidobacterium*
*longum*, *Candida rugosa*, and the components of kefir [56,57,58].

### 2.2. Parkinson’s Disease

Parkinson’s disease (PD) is the second most common neurodegenerative disease. This pathology is associated with progressive neuron loss and musculoskeletal dystrophy. PD is characterized by the accumulation of alpha-synuclein (α-syn) protein aggregates and an exponential increase in cell death [59]. α-syn stimulates microglial activation, leading to the death of dopaminergic neurons and the exacerbation of inflammation through the release of proinflammatory cytokines [60].

Since patients with PD have ID, it has been suggested that the GM may be a potential regulator of the pathogenesis of the disease [61]. In an animal model, PD development was associated with a low expression of bacteria belonging to the taxa *Bacteroides*, *Lactobacillus*, *Prevotella*, *Peptostreptococcus*, and *Butyricicoccus*, and a substantial increase in the genera *Enterobacter* and *Proteus* spp. [62,63]. In addition, a study of stool samples from PD patients found an increase in butyrate-producing species such as *Blautia*, *Coprococcus*, and *Roseburia*, as well as bacteria of the genus *Faecalibacterium*, compared with samples from healthy individuals [64].

Other studies have shown that patients with PD have significantly increased intestinal permeability, which is related to an increase in α-syn at the intestinal level [65]. ID has also been associated with the development of motor dysfunction and microglial activation, and these events were strongly associated with the concentration of SCFAs [66]. Other associations between changes in GM and PD include the overgrowth of pathogens such as *Helicobacter pylori* [67] and constant changes in the microbiota depending on the stage of development of the disease [62].

The use of probiotics and their metabolites has shown that the development of PD can be regulated, especially in its early stages. This is based on the regulation of SCFA levels in the gut [68]. Additionally, the use of a probiotic mix of *Lacticaseibacillus rhamnosum* (basonym of *Lactobacillus rhamnosus*), *L. acidophilus*, *Lactiplantibacillus plantarum* (basonym of *L. plantarum*), and *Enterococcus* in patients with PD was able to restore intestinal barrier integrity by strengthening the tight junctions. This approach also changed the bacterial composition of the GM, which was associated with a decrease in inflammatory components [69,70].

### 2.3. Autism Spectrum Disorder

Autism spectrum disorder (ASD) is an early neurodevelopmental disorder with cognitive features, including alterations in communication and social interaction, sensory abnormalities, repetitive behaviors, and variable levels of intellectual disability [71,72]. The etiology of ASD is complex and depends on various genetic and hereditary factors, from changes in chromosome number to maternal age, perinatal hypoxia, and factors related to diet and medication use during fetal pregnancy [61].

Some GI problems have been associated with ASD in children, including constipation, diarrhea, gastroesophageal reflux, and abdominal pain. These problems exacerbate the behavior of patients with ASD. Although it has not yet been proven, the cause of associated GI issues is thought to be a change in the composition of the GM. The main species associated with dysbiosis in ASD are *Bacteroides*, *Barnesiella*, *Odoribacter*, *Parabacteroides*, *Prevotella*, *Alistipes*, *Proteus*, *Shigella*, and *Parasutterella*, which are found in greater abundance in ASD patients. In contrast, the abundance of *Bifidobacterium* species is significantly reduced [61,73,74].

Both animal models and clinical trials have been used to investigate whether the restoration of the GM through the administration of probiotics has a beneficial effect on ASD development and treatment [75]. The results observed include products at the GM level, the GI, and behavioral symptoms [76,77,78,79,80,81,82,83,84,85]. Some of the studies are summarized in Table 1.

### 2.4. Multiple Sclerosis

Multiple sclerosis (MS) is an autoimmune and inflammatory disease of the CNS, characterized by the degradation of the myelin sheaths that cover the brain and spinal cord [87]. The MS etiology has not been fully elucidated, but the main cause is known to be the loss of self-tolerance developed against various CNS antigens and myelin sheaths [88,89].

This disease depends on both genetic and environmental factors, such as vitamin D deficiency, tobacco use, obesity, and infection with the Epstein–Barr virus [90]. In addition, several studies in both murine models and humans have demonstrated important differences between MS patients compared with healthy subjects; however, these alterations require further research since different characteristics have been found depending on the geographical region [91]. Within the alterations in the GM in MS, there is a lower abundance of Firmicutes and *Bifidobacterium*, as well as a decrease in the abundance of the *Lactobacillus* genus, especially *Limosilactobacillus reuteri* [91,92]. Bacteria such as *Methanobrevibacter*, *Escherichia*, and *Shigella* have been found in abundance in MS patients [49,93,94,95,96].

Given the relationship between ID and MS, attempts have been made to supplement classical treatments with probiotic administration [97]. Among the probiotics used, *L. casei*, *L. acidophilus*, *L. reuteri*, *B. bifidum*, and *Streptococcus thermophilus* have been the most successful [98,99]. A study tested the oral administration of a probiotic mix that included *Lactobacillus*, *Bifidobacterium*, and *Streptococcus* for 2 months in MS patients and reported a decrease in bacteria associated with MS, such as *Akkermansia* and *Blautia*, as well as an increase in the anti-inflammatory response, which suggests a synergistic effect of these probiotic strains with traditional therapy [100].

Other double-blind, placebo-controlled clinical studies have tested probiotics in the treatment of MS with promising results: (1) treatment with *L. acidophilus*, *L. casei*, *B. bifidum*, and *L. fermentum* for 12 weeks managed to improve general health and scores on the extended disability scale and Beck’s depression, anxiety, and stress scales, as well as beneficially regulate C-reactive protein (CRP), nitric oxide, and malondialdehyde metabolites [101]; (2) this probiotic mix was also shown to downregulate IL-8 and TNFα gene expression in MS patients [102]. These results were also replicated in other populations [103,104].

### 2.5. Depression and Anxiety

Depression and anxiety are two related disorders that represent serious problems for public health. These disorders are characterized by physical and emotional deterioration, as well as problems with social functioning [105]. These types of disorders can occur at any stage of life and are difficult to treat due to high relapse rates [106]. Depression is defined as “a common mental disorder that causes people to experience low mood, loss of interest or pleasure, feelings of guilt or low self-esteem, sleep or appetite disturbances, low energy and poor concentration” [107], while anxiety represents “a persistent feeling of discomfort, worry or fear” [108].

Anxiety and depression disorders are commonly associated with various alterations in the GM. It has been proposed that ID could regulate mental processes, such as mood, behavior, and memory [109].

The ID associated with these disorders has been verified in both depressed patients and nonhuman primate models [105]. In this regard, a marked difference was found in the GM composition of depressed patients, in which the bacterial populations belonging to Bacteroidetes were diminished, while the abundances of the phyla Actinobacteria and Firmicutes were notably increased [110,111,112]. In anxiety patients, there is an increase in Bacteroidaceae and a reduction in the phyla Firmicutes and Tenericutes [113]. Microbiota analyses in animal models of anxiety and depression have reported a decrease in the abundance of the Fusobacteria phylum and in the *Lactobacillus*, *Prevotella*, and *Bifidobacterium* genera, as well as a decrease in the Actinobacteria:Proteobacteria ratio [114,115].

There are a large number of preclinical and clinical trials in which probiotics are used for the treatment of behavioral patterns characteristic of depression and anxiety. Among these, the use of *B. longum* was evaluated in a double-blind, randomized, placebo-controlled trial in patients with mixed disorder, and an improvement was found, with a decrease in depressive traits [105]. In another study, *L. plantarum* was administered, and in addition to improving depressive behavior, an increase in learning abilities was also observed [116]. In another triple-blind, placebo-controlled study, a probiotic mix containing *B. bifidum*, *Bifidobacterium lactis*, *L. acidophilus*, *Levilactobacillus brevis*, *L. casei*, *Ligilactobacillus salivarius*, and *Lactococcus lactis* was administered to depressed patients, and the results showed a significant improvement in the symptoms of depression alongside reduced cognitive reactivity [117].

## 3. Microbiota–Gut–Skin Axis (MGS)

The skin, together with the mucosal epithelial barriers of the human body, is part of innate immunity and is the first defensive barrier against the external environment. It is essential in protecting against physical, chemical, and biological damage, preventing water and nutrient loss, regulating temperature, and participating in the immunological and neuroendocrine functions necessary for maintaining homeostasis [118,119].

The outermost layer of the skin is mainly responsible for its barrier function. The epidermis, in turn, consists of four cell layers in which stratum basalis stem cells differentiate into epidermal cells or keratinocytes of the stratum spinosum and stratum granulosum until they reach the stratum corneum, where the keratinization process converts them into corneocytes, cells without a nucleus that are keratinized and layered in a “brick and mortar” structure. In addition, keratinocytes express antimicrobial peptides, cytokines, and chemokines in the defense against pathogens. The epidermis also contains other immune cells, such as Langerhans cells. Additionally, both the epidermis and dermis contain dendritic cells, macrophages, mast cells, NK cells, CD4+ T, CD8+ T, and Treg lymphocytes [120].

When follicular structures such as hair follicles, eccrine and apocrine ducts, and sebaceous glands are considered, the skin has one of the most extensive epithelial surfaces, at 25 m^2^ [121]. As the mucous membranes of the GI, respiratory, and genitourinary tracts, the epithelial surface of the skin is in constant contact with millions of microorganisms, with which it maintains a symbiotic relationship through the epithelial cells and the immune system. This relationship is disturbed in dermatological diseases, such as acne, atopic dermatitis, psoriasis, and others, and dysbiosis may play an essential role in the etiology of these pathologies [122]. Moreover, it has been reported that not only the skin microbiota is altered in dermatoses but also the GM [119]. Similarly, a relationship between the skin and gut is observed in GI diseases in which cutaneous manifestations are observed [123]. This has led to the theory that there is a bidirectional communication axis between the skin and gut, involving the microbiota in contact with these epithelial surfaces, termed the “microbiota–gut–skin axis” [123,124].

The possible mechanisms that enable the link between the GM and the homeostatic or altered state of the skin are as follows: (1) metabolites produced by gut bacteria entering the bloodstream, (2) the translocation of gut bacteria, and (3) the modulation of the immune response by the GM [119]. Studies in mice have shown that the oral administration of probiotics such as *L*. *reuteri* led to an increase in dermis thickness, folliculogenesis, and the production of sebocytes, as evidenced by a shinier coat. These effects are only possible due to the production of the anti-inflammatory cytokine IL-10, as these differences were not observed in mice deficient in this cytokine [125]. In humans, the oral administration of *L*. *b**revis* SBC8803 resulted in improved hydration of the stratum corneum of the skin and decreased transepithelial water loss (TEWL) [126]. This suggests that the oral ingestion of probiotics benefits distal organs such as the skin through anti-inflammatory cells and cytokines to restore homeostasis, providing a potential alternative route for treating various skin diseases (Figure 2).

### 3.1. Atopic Dermatitis

Atopic dermatitis (AD), or eczema, is the most common chronic inflammatory dermatosis, occurring in 2–10% of adults and 15–30% of children [127,128]. It is characterized by severe pruritus in eczematous lesions that develop into crusty erosions and exudative or lichenified plaques, epidermal barrier disruption, dry skin, and IgE-mediated sensitization to food and environmental allergens [129]. Histologically, the plaques show epidermal intercellular edema and an infiltrate of lymphocytes, monocytes, macrophages, dendritic cells, and eosinophils [129]. One of the possible causes of barrier dysfunction is mutations in the gene encoding filaggrin, resulting in the loss of function of this protein, which plays a vital role in maintaining barrier function [127,128]. This increases susceptibility to invasion by environmental antigens, leading to an enhanced type 2 immune response, which compromises the integrity of the epidermal barrier.

The role of the microbiota in AD development has been explained by the microbiota hypothesis, which states that a modern lifestyle reduces exposure to microorganisms and alters the microbiota composition, which, in turn, leads to the incomplete development of the immune system and favors allergy development [130,131]. The establishment of ID preceding the development of this disease is evidence to support this theory, as decreased microbial diversity and colonization by pathogens such as *Escherichia coli* and *Clostridium difficile* at a young age is associated with increased AD risk in infants [132,133,134,135,136]. This dysbiosis continues with the development of the disease, as AD patients have higher levels of pathogenic bacteria such as bacteria of the genus *Clostridium*, *C. difficile*, *E. coli*, *Staphylococcus aureus*, and others, and lower levels of symbiotic bacteria such as *Bifidobacterium*, *Faecalibacterium prausnitzii*, and *Akkermansia muciniphila* [135,137,138,139,140].

ID in AD is enriched by proinflammatory metabolites that increase the proportion of IL-4-producing CD4+ T lymphocytes and reduce the proportion of Foxp3+CD25+CD4+ T lymphocytes [139]. Other studies have positively correlated the frequency of these pathogens with the eosinophil proportion and IgE level in the blood [132,138]. In addition, AD patients have low levels of SCFAs, such as butyrate and propionate, due to a reduction in the number of bacteria producing these metabolites, which is associated with AD severity [141,142]. These results suggest an essential role of the GM in the pathogenesis of AD, promoting the proinflammatory response and affecting the epithelial barrier’s function since SCFAs are necessary for its integrity and have an anti-inflammatory effect. The reduction in the number of bacteria producing these metabolites and the increase in pathogens lead to a deterioration of intestinal and epidermal barrier function and exacerbate the clinical manifestations of AD.

There is evidence that probiotic treatment reduces the severity of the clinical manifestations of AD [143], although further studies are needed to confirm this since others report no effect [144]. In addition, several studies have reported the potential use of oral probiotics to prevent this condition [145,146]. Probiotic mixtures containing *Bifidobacterium* and *Lactobacillus* are the most commonly used in these studies. Various strains, such as *B. bifidum*, *B. longum*, *Bifidobacterium breve*, *L. rhamnosus*, *L. acidophilus*, and others, administered during pregnancy and early infancy in women with a family history of this disease, reduce the risk of developing AD in infants [147,148,149,150,151].

Regarding the effect of oral probiotics in treating AD, studies in mice have reported a reduction in AD lesions, erythema, scratching, and epidermal thickness [152,153,154,155]. Various strains of *L. rhamnosus*, *L. plantarum*, *F. prausnitzii*, and *A. muciniphila* reduce infiltration by eosinophils and mast cells, reduce the levels of IgE and cytokines associated with a Th2 profile, such as IL-4 and IL-5, and reduce the levels of cytokines that induce this lymphocyte subtype, such as TSLP (thymic stromal lymphopoietin). The administration of these probiotics increases anti-inflammatory cytokines, such as IL-10, and cytokines with a Th1 profile, such as IFN-γ [152,153,154,155]. These changes in the inflammatory profile are induced through the modulation of the immune system by probiotics to restore the balance of Th1/Th2 lymphocytes via an increase in Treg lymphocytes [144], in addition to improving the function of the intestinal and epidermal barrier by increasing the production of filaggrin in the skin and the tight junction proteins ZO-1 and claudin-1 in the intestine [155].

A recent study by Fang et al., in which they used *B. longum* CCFM1029 in a mouse model of AD, postulated another mechanism of action of probiotics. This probiotic modulates the GM to increase the production of indole-3-carboxaldehyde, a tryptophan metabolite, which suppresses the Th2 response via the aryl hydrocarbon receptor (AHR). AHR activates signaling pathways that inhibit the transcription factor for differentiation to Th2, STAT6 [156]. Similarly, *B. longum* inhibits the expression of TSLP, which promotes the differentiation of these lymphocytes and is involved in the pathogenesis of AD [156,157].

Finally, in humans, an improvement in clinical symptoms in children and adults with AD has been reported [158,159,160,161]. The ingestion of probiotics from *Bifidobacterium* and *Lactobacillus* species is associated with a reduction in AD severity and TEWL, as well as an improvement in skin hydration [158,159,160,161]. These studies report that this improvement is related to the modulation of the GM to favor tryptophan metabolism, the reduction in microbial translocation, the increase in Treg and Th1 lymphocytes, and the decrease in Th2 lymphocytes, IgE, and proinflammatory cytokines such as TNF, TSLP and the chemokine CCL17 [156,158,159,160]. Table 2 shows a summary of relevant and recent studies about the use of probiotics in AD.

### 3.2. Psoriasis

Psoriasis is a chronic inflammatory disease of the skin with a prevalence of 0.91–8.5%, which varies widely depending on the geographic region, occurring between the ages of 15 and 25 years [163,164]. It is characterized by thickened, inflamed, and scaly erythematous lesions, often accompanied by well-defined plaques, with pruritus in 50% of cases, which can occur anywhere on the body [164]. These clinical manifestations are the result of the hyperproliferation of keratinocytes with incomplete keratinization, accompanied by an inflammatory infiltrate composed of dendritic cells, macrophages, T lymphocytes, and neutrophils [165]. Etiology and pathogenesis are still not well understood, although genetic, environmental, and immunological factors influence the development of the disease [164].

Abnormal intestinal structures with a smaller epithelial surface and loss of intestinal integrity have been reported in patients with psoriasis [166,167], as well as an increased risk of developing inflammatory bowel diseases (IBD) such as ulcerative colitis, Crohn’s disease, and celiac disease [168]. The GM of patients with psoriasis has a similar microbial composition to those of people with IBD and differs from the microbiota of healthy people [169,170]. Analysis of the GM of these patients shows a decrease in the abundance of Bacteroidetes and an increase in Firmicutes [169,171], as well as a reduction in symbiotic bacteria such as *F. prausnitzii*, *A. muciniphila*, and *Prevotella copri* with a concomitant increase in pathogenic bacteria such as *E. coli* [169,170,172]. These results indicate a possible relationship between gut dysbiosis and psoriasis that should be explored.

Studies in mice and humans show how probiotics help to reduce the levels of proinflammatory biomarkers and the characteristic lesions of the disease. Chen et al., reported that the consumption of *Lactobacillus pentosus* GMNL-77 in mice with imiquimod-induced psoriasis resulted in less erythema, scaling, and epidermal thickening, as well as a reduction in the expression of proinflammatory cytokines such as TNF-α, IL-6, IL-23, IL-17A, IL-17F, and IL-26 [173]. In humans, two studies reported that the consumption of *Bifobacterium infantis* 35624, *B. longum* CECT 7347, *B. lactis* CECT 8145, and *L. rhamnosus* CECT 8361 helped reduce plasma levels of CRP and TNF-α, modulated the GM with a significantly higher number of beneficial genera, and were also associated with a clinical improvement in lesion severity in psoriasis patients [174,175]. Probiotics thus show promise in the treatment of psoriasis.

### 3.3. Acne

Acne vulgaris is a chronic inflammatory disease of the pilosebaceous unit characterized by comedones, small bumps caused by follicular obstruction and sebaceous gland hyperplasia, and painful inflammatory lesions such as papules, pustules, and cysts [176,177,178]. Worldwide, it is the eighth most common disease, accounting for 0.29% of the global disease burden, and the second most common dermatologic disease [179,180]. In developed countries, it affects 85–90% of people aged between 12 and 24 years [176,181].

It is thought to be a multifactorial disease caused by the oversecretion of sebum, the obstruction of the excretory ducts due to the abnormal desquamation of keratinocytes, and the proliferation and colonization of *Cutibacterium acnes*, which secretes proinflammatory mediators in the comedo [177,178]. In addition, a diet with a high glycemic index and a high level of milk protein plays an essential role in pathogenesis by increasing insulin signaling and insulin-like growth factor-1 (IGF-1), which inhibits the transcription factor that regulates mTORC1, Fox01. The activation of mTORC1 leads to hyperproliferation, lipogenesis, and keratinocyte hyperplasia, contributing to the development of acne [182].

Several studies have shown that the GM of patients with acne differs from that of healthy controls. Acne patients have lower microbial diversity, a lower abundance of Firmicutes, and an increase in Bacteroidetes [176,183,184]. These differences are gender-dependent, as male acne patients have higher dysbiosis levels than females [183]. Similarly, a decrease in the number of bacterial phyla with anti-inflammatory properties and producers of metabolites, such as SCFAs, with antimicrobial and immunomodulatory activity has been reported. These include *Bifidobacterium*, *Butyricicoccus*, *Lactobacillus*, *Lactococcus*, *Clostridium*, Clostridiales, Lachnospiraceae, and *Ruminococcus* [176,183,184]. This suggests that patients with acne have ID, which may exacerbate the skin manifestations of this pathology.

Orally administered probiotics represent an alternative for treating acne that has shown promising results, although further studies are needed. In 1961, the first study was published in which the administration of the probiotics *L. acidophilus* and *Lactobacillus delbrueckii* subsp. *bulgaricus* resulted in clinical improvement in patients with acne, especially in cases with inflammatory lesions [185]. Jung et al., showed that synergistic treatment with the probiotics *L. acidophilus*, *L. bulgaricus*, and *B. bifidum* and the antibiotic minocycline resulted in a more significant decrease in acne skin lesions than treatment with the antibiotic alone [186]. A study by Fabbrocini et al., showed that the administration of *L. rhamnosus* SP1 to acne patients also improved skin condition, with a reduction in lesions due to a decrease in the expression of IGF-1 and an increased expression of Fox01 [178]. Thus, the beneficial effects of these particular probiotics arise from the regulation of IGF-1/mTORC1 metabolism and the increased expression of anti-inflammatory cytokines and molecules that affect the microbiota–gut–skin axis, reducing the exacerbated inflammation found in acne.

## 4. Microbiota–Gut–Lung Axis (MGL)

The microbiota–gut–lung axis refers to the connection between the two systems, mainly due to the ability of the GM to influence the course and outcome of lung disease and vice versa [187]. The commensal microbiota is established in the mucous membrane tissues exposed to the external environment, including the lung and gut [188]. Histologically, there are four layers in the respiratory and GI tracts: mucosa (epithelium and lamina propria), submucosa, cartilage and muscle, and adventitia [189]. The epithelium is the primary physical protective barrier in the respiratory and GI tracts against components of the vascular system and the intestinal or airway lumen.

The respiratory system interacts with pathogens, allergens, and other particles through air inhalation and exhalation [189,190]. In the lower respiratory tract, alveolar macrophages and various subsets of dendritic cells act as sensors, as they warn of the presence of pathogenic bacteria, viruses, or fungi through pattern-recognition receptors. On the other hand, tissue-resident lymphocytes such as innate lymphoid cells, NK cells, natural killer T (NKT) cells, mucosa-associated invariant T (MAIT) cells, epithelial γδ-T cells, and tissue-resident memory T cells are also found. Together, they trigger the mechanisms required for timely pathogen elimination or tolerance of the resident and transient microbiota [191].

The microbiota of the respiratory system is very different from that of the gut [192]. The lung microbiota composition depends on several factors, such as bacteria immigration and excretion through the airways, as well as the relative growth rate in the pulmonary mucosa [193]. In healthy subjects, the central lung microbiota consists of *Pseudomonas*, *Streptococcus*, *Prevotella*, and *Fusobacterium*, followed by *Haemophilus*, *Veillonella*, and *Phorphyromonas*, with *Pseudomonas* as the dominating genus of the three lobes of the lung [194]. The Pulmonary HIV Microbiome Project, one of the most extensive multicenter studies, characterized the lung and airway microbiome from 2009 to 2015, revealing that the most abundant bacterial genera in the mouth and lungs were *Streptococcus*, *Prevotella*, and *Veillonella*, while in the lungs the following taxa were overrepresented: *Haemophilus*, Enterobacteriaceae, *Methylobacterium*, and *Tropheryma* [195].

The intestines and lungs communicate through lymphatic and blood circulation. The largest lymphatic vessel is the thoracic duct, and the lymph from the thoracic duct enters the left subclavian vein, which means that the lung is the first organ to encounter the mesenteric lymph [196]. For this reason, the gut–lung axis is regulated mainly by the GM, as well as the immune response it induces [197,198]. To date, several studies have demonstrated the involvement of gut dysbiosis in bidirectional pulmonary complications via the microbiota–gut–lung axis. Probiotics and prebiotics used for the treatment of different pulmonary diseases are shown in Table 3.

### 4.1. COPD

Chronic obstructive pulmonary disease (COPD) is the third leading cause of death worldwide [210]. It is considered a chronic degenerative disease characterized by persistent respiratory symptoms, airflow limitation, and the development of emphysema. The main etiologic factors are smoking and exposure to harmful gasses or particles, in combination with genetic factors [211].

Other tissues are affected in addition to the respiratory tract in patients with COPD. Rutten et al., demonstrated for the first time that enterocyte damage and increased intestinal permeability in the intestine are due to a mismatch of the ventilation/perfusion ratio, which leads to tissue hypoxia [212]. Kirschner et al., reported the loss of integrity in the small intestine barrier in COPD patients, with increased intestinal permeability in active and former smokers. They also found a decrease in the plasma concentration of some SCFAs, mainly acetate. While the concentration of SCFAs is not associated with other parameters of intestinal integrity, it suggests a possible alteration in the GM and its ability to produce these metabolites [213].

Li et al., elucidated some of the underlying mechanisms of gut dysbiosis and its importance in disease progression. The results show that patients with complications had a lower abundance of bacteria of the families Bacteroidaceae and Fusobacteriaceae compared with healthy subjects. This was also associated with a significant decrease in the total SCFA concentration in stool samples. To support this association, mice that received fecal microbiota transplantation from COPD patients showed lung inflammation, emphysema, airway remodeling, mucus hypersecretion, and impaired lung function [214].

The damage caused to the intestinal barrier integrity and the ID contribute to the migration of metabolites, cytokines, and cells residing in the gut to the lung tissue, where they act as effectors and further contribute to systemic inflammation. In this context, Wang et al., demonstrated that GM restoration improves pulmonary inflammatory parameters in mice with COPD. They also describe the dominating bacterial genera in COPD mice as being *Candidatus_Stoquefichus*, *Streptococcus*, and *Marvinbryantia*, which are negatively associated with body weight and lung function and positively associated with Th17/Treg balance and proinflammatory cytokine concentration [215]. In another study in COPD patients, an association between the GM and the decrease in forced expiratory volume in one second (FEV1) was found. Firmicutes and *Stenotrophomonas* were increased in the more severely affected patients, whereas control subjects had increased Bacteroidetes, *Alloprevotella*, and *Acinetobacter*. These results show that, as concluded by the authors, “pulmonary inflammatory status in mice and patients with COPD may be modulated in part by their GM” [216] (Figure 3).

Smoking and poor diet are important risk factors for COPD development and ID. In COPD mice, supplementation with *L. rhamnosus* and *B. breve* prevents airway inflammation and lung injury, reduces bronchoalveolar lavage, and restores the cytokine and chemokine balance [217]. The effect of dietary fiber on this disease has also been studied in mouse models. Jang et al., demonstrated that a high-fiber diet attenuated the physiological changes associated with emphysema progression and the inflammatory response. They also emphasized that dietary fiber modulates microbial diversity and its metabolites, such as SCFAs, bile acids, and sphingolipids [200].

On the other hand, in a randomized, double-blind trial with humans, Koning et al., investigated whether the intake of multistrain probiotics during and after antibiotic treatment alters the GM of COPD patients. However, no differences in GM composition were found [218].

### 4.2. Asthma

Asthma is a chronic respiratory disease that affects between 1 and 18% of the population in different countries, with symptoms such as wheezing, shortness of breath, chest tightness, coughing, and variable expiratory airflow limitation. It is associated with airway hyperresponsiveness and inflammation [219]. Asthma is a complex, multifactorial, and heterogeneous respiratory disease [220].

There is still no conclusive evidence of a causal relationship between asthma and the gut microbiome. However, asthma was the second most common comorbidity in an IBD cohort [221]. A meta-analysis described 15 studies that reported an association between asthma and Crohn’s disease, and 16 studies showed an association with ulcerative colitis. In most of these studies, IBD preceded the onset of asthma [222]. An analysis of 456,327 Europeans showed that the presence of asthma in childhood increased the risk of developing gastroesophageal reflux, irritable bowel syndrome (IBS), and peptic ulcer disease [223]. Shen et al., conducted two retrospective cohort studies, one with asthma patients and the other with IBS patients. They found that asthma patients were at significantly higher risk for IBS and vice versa [224]. However, to date, there is no conclusive evidence to suggest that both conditions arise due to a similar pathology and risk factors [222].

Environmental risk factors for asthma development include the role of the GM via the microbiota–gut–lung axis. Differences have been found in the GM between healthy subjects and patients with asthma. In a population from Guangzhou, China, specifically at the taxonomic family level, a higher relative abundance of Veillonellaceae and Prevotellaceae was observed in patients with asthma, in addition to a decrease in Acidaminococcaceae, which positively correlates with the mean FEV1 and disease severity [225].

In an asthma murine model, the attenuation of physiological and histological asthmatic features increased the relative abundance of the phylum Verrucomicrobia and of *A. muciniphila* in the lung microbiota. In addition, an increase in tight junction protein expression in alveolar epithelial cells and a decrease in LPS biosynthesis and mucin production were observed. This, in turn, led to an increase in the abundance of beneficial bacterial species in the gut, such as *Bacteroides acidifaciens*, and, consequently, an increase in butyric acid concentration [226] (Figure 3).

A study in asthma patients classified by allergic and nonallergic phenotypes in southern China found that the gut microbiome profile differed significantly between healthy and unhealthy subjects and between asthma phenotypes. In this context, it was found that the disease state and serum IL-4 concentration were the two main contributors to the differences in gut microbiome between patients and controls. The findings include 28 different bacterial species between controls and patients and 17 species with diagnostic potential in the stratification of asthma with or without allergy. These data suggest a close relationship between the gut microbiome and differences in disease state [227].

Because asthma develops early in human life and depends on prenatal and postnatal stages of pregnancy, probiotic supplementation is usually investigated during this period. Murine models have been used to demonstrate promising therapeutic approaches, such as *L. lactis* NZ9000, which favored a decrease in leukocyte inflammation. *L. rhamnosus* GR-1 decreased airway hyperresponsiveness and contributed to a reduction in asthmatic disorders. *B. breve* M-16V promoted a decrease in eosinophils in the bronchoalveolar fluid of newborn mice, thus reducing lung inflammation due to allergies caused by environmental pollution [204,205]. Mice with asthma treated with *L. reuteri* ATCC 23272, *L. rhamnosus* GG, or *B. lactis* Bb-12 showed decreased airway hyperreactivity and reduced bronchoalveolar cell inflammation [228]. In humans, the results are somewhat more variable. In children with eczema, asthma, and allergic rhinitis, *L. rhamnosus* GG was administered. However, no reduction in the risk of asthma development was observed, but the wheezing frequency was lower, so probiotics are not considered a therapeutic agent for this disease [207].

Postbiotics such as vitamin D display an essential role as a modulator of the intestinal epithelial barrier, as well as the homeostasis of the microbiota. The Vitamin D Antenatal Asthma Reduction Trial (VDAART), conducted by Lituonja et al., investigated whether vitamin D supplementation in pregnant women can prevent asthma and allergy development in their children. It was found that vitamin D supplementation in pregnant mothers did not alter the risk of asthma in their children [229]. Other metabolites such as SCFAs have also been studied. In a prospective study that included individuals with and without bronchial asthma, those with higher concentrations of butyrate and propionate showed an increase in T cell differentiation, reducing allergic reactions [230]. The differences between the results obtained in mice and humans may be attributed to methodological differences, selection criteria, and differences in environmental regulation between mice and humans.

### 4.3. COVID-19

Severe acute respiratory syndrome coronavirus 2 (SARS-CoV-2) is the cause of coronavirus disease 2019 (COVID-19); its genome is composed of positive single-stranded RNA ssRNA (+), and it belongs to the order Nidovirales, family Coronaviridae, and species SARS [231]. Transmission occurs by droplets from people with respiratory symptoms and by contact with surfaces contaminated with a viral load [232,233,234]. The first case worldwide was reported as “pneumonia of unknown cause” on 31 December 2019, in Wuhan, China [235]. Since then, the virus has spread rapidly. In March 2020, the World Health Organization (WHO) declared COVID-19 a pandemic. It currently represents the most significant public health problem, with 539,119,771 confirmed cases worldwide (as of 23 June 2020), of which 6,322,311 have died [236].

COVID-19 primarily manifests as a respiratory infection, with the most reported symptoms being fever, fatigue, and dry cough. Pneumonia and cytokine storm are typical clinical manifestations of COVID-19 [237]. Cells susceptible to SARS-CoV-2 infection include respiratory epithelial cells, alveolar macrophages, intestinal epithelial cells, myocardiocytes, olfactory cells, bile duct cells, and Sertoli testicular cells. They all exhibit high expression of the ACE2 receptor, which is necessary for recognizing and integrating the virus into host cells. Therefore, infection with this virus leads to various respiratory diseases, heart failure, and GI symptoms, such as diarrhea, vomiting, and nausea [238].

As for damage to the GI tract, SARS-CoV-2 could impair tryptophan absorption by binding to ACE2 in the intestine [239]. According to work by Hashimoto et al., this leads to the decreased secretion of antimicrobial peptides in the gut [240], which has been shown to affect microbiota composition and promote ID [241]. Patients with niacin or tryptophan deficiency are known to suffer from pellagra, and 90% of them develop severe colitis and diarrhea, conditions also associated with COVID-19, either as a complication or as a risk factor [242,243].

To date, it is unclear whether the GI symptoms of patients with COVID-19 result from fecal–oral infection or are due to indirect mechanisms mediated by an association with the pulmonary mucosa [196,244]. In SARS-CoV-2 infection, local immune cells produce proinflammatory cytokines, which affect extrapulmonary tissues such as the GI tract. Then, they activate resident cells, and this inflammatory response is associated with dysbiosis and increased intestinal permeability, promoting systemic and pulmonary inflammation [245].

Several papers have linked the GM to COVID-19 severity based on the immunomodulatory capacity of local cells that emit systemic signals [246,247,248]. Tang et al., demonstrated the association between the abundance of certain intestinal bacteria and clinical indicators in patients with COVID-19 pneumonia. The relative abundance of probiotic (*Lactobacillus* and *Bifidobacterium*) and anti-inflammatory bacteria (*F. prausnitzii*, *Clostridium butyricum*, *Clostridium leptum*, and *Eubacterium rectale*) was negatively associated with proinflammatory markers in serum (IL-6, CRP, and neutrophil level), indicators of liver damage (ALT and AST), and markers of organic dysfunction (D-dimer, LDH, and creatine kinase) (Figure 3). On the other hand, bacteria known for their pathogenic effect (*Enterococcus*, Enterobacteriaceae, and *Atopobium*) correlated positively with these indicators. In a group of critical patients, the GM is enriched in pathogenic bacteria [249]. In a cohort study, the GM in patients positive for SARS-CoV-2 correlated with an increase in the concentration of proinflammatory cytokines and other inflammatory markers, such as CRP, LDH, AST, and GGT. The relative abundance of *F. prausnitzii*, *E. rectale*, and *Bifidobacterium* species is decreased in stool samples from patients even 30 days after infection, which the authors believe may contribute to the late symptoms of the disease [245].

Although there are several studies describing the ability of probiotics, prebiotics, and postbiotics to reduce clinical severity in other respiratory diseases, for COVID-19, there are not yet any conclusive results [250]. A study by Ceccarelli et al., consisting of a cohort of patients who tested positive for SARS-CoV-2, characterized their GM. Those who were symptomatic for a more extended period showed more severe dysbiosis. These individuals received multiple supplements with *Lactobacillus helveticus*, *L. acidophilus*, *L. paracasei*, *L. plantarum*, *L. brevis*, *S. thermophilus*, and *B. lactis*. In patients with pneumonia, those who received this supplementation had a higher survival rate [208]. On the other hand, the use of yogurt as a support food and supplement for probiotics and postbiotics, with micronutrients such as zinc, potassium, magnesium, and calcium, has been suggested. This suggestion is mainly because active peptides from casein are present in yogurt that act as inhibitors of ACE2 expression, thus reducing viral entry into host cells [251]. Furthermore, bioinformatics tools were used to determine which probiotics and bacterial strains can reduce the expression of the genes ACE, AGTR1, and ACE2. As a result, it was found that *Bacteroides thetaiotaomicron*, *Bacteroides fragilis*, *A. muciniphila*, and *F. prausnitzii* beneficially modulated miR-124-3p and miR-26b-5p miRNAs, which modified the expression of these genes [252]. Although bioinformatic analyses are very descriptive, they can never replace in vitro and in vivo assays that allow us to see the interaction and response under normal patient conditions.

## 5. Microbiota–Gut–Heart Axis (MGH)

The GM and its metabolites are associated with cardiovascular disease (CVD) progression, including hypertension, dyslipidemia, atherosclerosis, thrombosis, heart failure, and ischemic stroke [253]. In addition, the GM influences the response to cancer therapy and susceptibility to toxic side effects [254]. Drugs can alter the microbiome and cause side effects that are independent of the drug molecule itself [255]. Recently, antitumor chemotherapeutic agents such as cisplatin and doxorubicin in the GM have caused disorders involved in the pathogenesis of cardiotoxicity [256,257]. Table 4 shows recent relevant studies on probiotic use in CVD.

### 5.1. Hypertension

Hypertension is the most common modifiable risk factor for CVD, the leading cause of death worldwide; although it is modifiable, it is still one of the most important. Excessive salt intake is associated with elevated blood pressure (BP), whereas a low-sodium diet lowers BP and reduces morbidity and mortality from CVD [266,267].

Although few studies have linked the GM to human hypertension, Joe et al., recently showed that germ-free rats have hypotension and reduced vascular contractility. Restoring the GM in these rats also restores BP and vascular contractility [268]. A study by Li et al., reported that the transplantation of the fecal microbiota of hypertensive patients to germ-free rats is associated with increased BP in these animals [269]. These results suggest a role for the GM in regulating BP.

Recent evidence shows that gut dysbiosis is associated with the development of hypertension [270,271]. This dysbiosis is highlighted by an increased ratio of Firmicutes to Bacteroidetes and a decrease in SCFA-producing bacterial species in murine models. At the same time, lower microbial richness is found in hypertensive patients [269,272].

In addition, analysis of the GM in patients with hypertension and prehypertension shows similarities in their composition, as well as an increased abundance of bacteria such as *Prevotella* and *Klebsiella* and a decrease in bacteria of the genera *Faecalibacterium*, *Oscillibacter*, *Roseburia*, *Bifidobacterium*, *Coprococcus*, and *Butyrivibrio* compared with healthy control subjects. The microbiome analysis shows a decrease in the activity of genes related to amino acid synthesis, fatty acid utilization, and saccharide transport. At the same time, the biosynthesis of metabolites such as LPS is increased [269]. This may indicate how gut dysbiosis causes low-grade inflammation, which may be involved in the development of hypertension.

Other bacterial metabolites associated with BP include trimethylamine (TMA) and trimethylamine-N-oxide (TMAO), which are gut microbial products derived from certain food components. TMAO is associated with CVD and CNS disease [273,274]. Elevated circulating TMAO induces an inflammatory response and oxidative stress not only in peripheral tissues, including heart, aorta, and kidney [275,276,277], but also in the CNS [278,279], as TMAO can rapidly overcome the BBB [280] (Figure 4).

Recent studies report that high salt intake alters the composition of the GM, which, in turn, increases the plasma concentration of TMAO in animals [271,281]. Recently, Liu et al., showed that long-term intake of a high-salt diet (HS) leads to an increase in the metabolite TMAO, produced by the GM, in the bloodstream and brain, which is associated with an increase in neuroinflammation and oxidative stress in the paraventricular nucleus (PVN). The inhibition of TMAO formation ameliorates HS-induced sympathetic arousal and hypertension by reducing neuroinflammation and oxidative stress in the PVN. These findings may provide new insights into HS-induced hypertension mechanisms [282].

The team of Wang et al., showed that using a small-molecule inhibitor of microbial choline TMA lyase activity suppressed microbial TMA and TMAO formation, macrophage foam cell formation, and atherosclerosis in vivo. Whether targeting this pathway leads to a parallel reduction in CVD risk in humans is still unknown. However, it is an essential area for future research [283].

On the other hand, beneficial metabolites produced by the GM, such as SCFAs, are also associated with BP because a reduction in the levels of these metabolites is usually associated with hypertension [272,284]. A study by Marques et al., showed that a diet high in fiber and acetate in hypertensive mice not only altered the GM and increased the proportion of acetate-producing bacteria but was also associated with a reduction in BP, cardiac fibrosis, and left ventricular hypertrophy [285], demonstrating that SCFAs are associated with low BP.

Finally, probiotic intake is also associated with improved BP in hypertensive patients. These effects are related to the duration and dose of probiotic treatment and the patient’s age. They are also related to other benefits such as lowering BMI and blood glucose levels [286].

### 5.2. Atherosclerosis

Atherosclerosis (AS) refers to plaques formed in arteries with a lipid core or atheroma surrounded by a fibrous layer, which is the underlying pathology of CVD, and its prevalence is increasing worldwide [287]. These atherosclerotic plaques contain bacterial DNA, and the bacterial taxa observed in the atherosclerotic plaques were also found in the intestines of the same individuals [288,289]. These observations suggest that the microbial communities at these sites may be a source of bacteria in plaque that could influence plaque stability and CVD development.

Recent studies report that AS is closely associated with abnormal chronic low-level inflammation and gut dysbiosis. Alterations in the microbiota are associated with inflammatory status and progression from AS to CVD in both mouse models and human patients [290,291,292,293].

Polyunsaturated fatty acids (PUFAs) can modulate the GM and inflammation and have pleiotropic benefits in chronic metabolic diseases [294]. However, the effects of dietary PUFAs on AS and their mechanisms have not yet been fully elucidated. Yiwei et al., conducted an interesting study investigating the effects of PUFAs in a mouse model of AS with apolipoprotein E (ApoE)^−/−^ deficiency. The results show that administration of flaxseed oil (FO), which is rich in α-linolenic acid (ALA), a PUFA, improved AS injury, body weight, and levels of bile acids, chronic systemic and vascular inflammatory cytokines, and macrophages. In addition, FO decreased LPS levels, improved intestinal integrity, and modulated GM and SCFAs. Treatment with FOs in combination with antibiotics reduced their beneficial effects, suggesting that the GM acts in synergy with PUFAs [295]. These results indicate that an FO-rich diet improves AS in ApoE^−/−^ mice via the microbiota–gut–heart axis.

The use of probiotics in AS appears to be protective, although the mechanism is not yet understood. Probiotics are associated with a reduction in AS lesions and inflammation in mouse models and AS patients [258,259,260]. However, this effect depends on the probiotics administered, as not all studies report these benefits [261].

### 5.3. Coronary Artery Disease

Coronary artery disease (CAD) is a significant cause of mortality in the world. Unstable angina (UA) is characterized by a set of angina symptoms typical of ischemic cardiovascular and cerebrovascular disease [296]. IGF-1 has been identified as a valuable biomarker and a therapeutic target for CAD. Clinical studies have shown that low circulating IGF-1 levels are highly associated with a high incidence of CAD [297,298]. IGF-1 has several beneficial effects, including reducing inflammation, reducing apoptosis, and the stimulation of angiogenesis, all of which are related to vascular function and AS [299].

Recent studies have highlighted the role of the GM between UA occurrence and development [300]. For example, a clinical study showed that UA patients had a lower abundance of Bacteroidetes and a higher abundance of Firmicutes than healthy individuals [301]. In addition, another multiomics analysis study demonstrated the complex interplay of the GM, circulating metabolites, and severity of UA [301].

It has been reported that the GM is involved in regulating IGF-1-related signaling and contributes to UA development. Although the mechanism of IGF-1 induction by the GM is unclear, metabolites such as SCFAs contribute to increased IGF-1 production [302].

Langsha Liu and Fanyan Luo conducted a comparative study between UA patients and healthy subjects, in which they found that the relative abundance of Bacteroidetes increased in UA patients. Some bacteria belonging to the Bacteroidetes species are involved in amino acid biosynthesis and degradation, suggesting that an imbalanced amino acid metabolism in UA patients may be due to its increased abundance. They also found a negative correlation between the relative abundance of Bacteroidetes and serum IGFBP-4 levels, suggesting that these bacteria may be related to the IGF-1 system and even to UA development. Another important finding of this study was that UA patients had a higher relative frequency of Synergistetes, which are associated with periodontal disease [303].

Because probiotics reduce inflammatory markers, they represent a potential avenue for CAD prevention. Studies in patients suffering from or at risk of developing CAD show how probiotics such as *L. plantarum* 299v reduce the risk of this pathology by decreasing BP and levels of proinflammatory cytokines such as IL-6, IL-8, and IL-12 and adipokines such as leptin and increasing SCFA levels [262,263]. Further studies are needed to determine which types of probiotics are most effective in this pathology, but these initial studies suggest that it is possible to improve the symptoms of CAD to prevent the exacerbation of the disease.

### 5.4. Heart Failure

Heart failure is a chronic and progressive disease caused by abnormal changes in cardiac structure and function [304]. There is a growing body of scientific evidence supporting the role of the gut in the development of heart failure, the so-called “gut hypothesis of heart failure.” This hypothesis states that decreased cardiac output and increased systemic congestion may cause ischemia and edema of the intestinal mucosa, leading to increased bacterial translocation and an increase in circulating endotoxins, which may contribute to the underlying inflammation observed in patients with heart failure [305].

Niebauer et al., found that patients with heart failure who had peripheral edema had higher plasma concentrations of endotoxin and inflammatory cytokines than patients without edema. After short-term diuretic therapy, serum concentrations of endotoxin, but not cytokines, decreased [306].

Another study in patients with heart failure showed that lower intestinal perfusion was associated with higher serum concentrations of anti-LPS IgA, which correlated with greater growth of bacteria obtained from biopsies of the intestinal mucosa but not from fecal bacteria. The nature of the bacterial microbiota also appeared to differ in these subjects from that of the control subjects [307]. Recently, Pasini et al., reported an increase in the amount of pathogenic intestinal bacteria such as *Campylobacter*, *Shigella*, and *Salmonella*, as well as *Candida* fungi, in patients with chronic heart failure compared with healthy control subjects [308]. These data suggest that an assessment of gut barrier function may lead to a better understanding of the effects of therapy on heart failure.

There is scientific evidence of the relationship between the GM and cardiovascular and metabolic function. Heart disease can alter the GM’s richness, diversity, and composition. Anwar et al., reported that a rat model of cardiac remodeling induced by the forced swimming-induced stress (FSIS) model resulted in gut dysbiosis with a reduction in microbial diversity compared with control animals. Vitamin C treatment decreased the abundance of Bacteroidetes while the abundance of spirochetes increased. Decreased CRP and creatine kinase myocardial band were also observed. The results suggest that FSIS-induced cardiac complications are also associated with changes in gut microbial abundance. Higher doses of vitamin C boost immunity by modulating the GM, which has a positive effect on the heart [309].

Although probiotics are considered an alternative treatment for various diseases, more studies are needed to determine their benefits in heart failure. In a mouse model of coronary artery occlusion, the use of the probiotic *L. rhamnosus* GR-1 shows attenuation of left ventricular hypertrophy and an improvement in ventricular function, suggesting that the probiotic alters disease progression to heart failure [264]. In another study conducted on patients with heart failure, administration of the fungal probiotic *Saccharomyces boulardii* lowered levels of inflammatory biomarkers such as creatinine, uric acid, and high-sensitivity CRP and improved cardiovascular function by decreasing the left atrial diameter and improving the left ventricular ejection fraction compared with the placebo group [265].

## 6. Relationship of Microbiota with Other Metabolic Alterations

Currently, obesity and type 2 diabetes mellitus (T2DM) are pathologies that are increasing relative to the total world population. Recent studies have identified ID as a key factor related to the metabolism of these diseases, with bacteria being able to produce metabolites that stimulate hormones and a proinflammatory immune response. Likewise, the low-grade inflammation found in these pathologies alters the microbial communities found in the gut, leading to exacerbation of the inflammation. The bidirectional interaction between the GM and metabolism shows a possible pathway to prevent and control metabolic diseases. While more studies are needed, these interactions hint that probiotic implementation in the treatment of obesity and T2DM is a viable pathway [310]. Table 5 summarizes the most relevant results about probiotic use from studies of obesity and T2DM.

### 6.1. Obesity

Globally, it is estimated that nearly 1.4 billion adults are overweight, while 500 million fall into the obese category [320]. Overweight and obesity are known inflammatory diseases that involve the abnormal accumulation of visceral adipose tissue. This adipose tissue influences the release of proinflammatory mediators such as TNF-α, IL-1β, and IL-6. These cytokines can act in endocrine or paracrine pathways and interfere with insulin signaling through the activation of the protein complex NF-κB [321,322].

The inflammatory state of obesity is not accompanied by an infection or by signs of autoimmunity and does not show signs of tissue damage; this is why some researchers use the term metainflammation, which refers to an inflammation state that is metabolically triggered [323]. Some studies have confirmed the association between the indexes of overweight and obesity with inflammatory markers such as CRP [324].

In addition, these pathologies are related to other nontransmissible chronic diseases, such as diabetes mellitus and cardiopathies [325]; however, there are no characteristic symptoms beyond abdominal growth [326]. These pathologies are commonly caused by an imbalance between dietary intake and energy expenditure [327,328], as well as other environmental and biological factors, such as an alteration in the GM, which plays an important role in the inflammation associated with these diseases [329].

The GM participates in the regulation of adipose tissue, and depending on its composition, the effect can be negative through the release of LPS, which promotes a proinflammatory immune response in the adipose tissue, or positive through SCFAs, which are associated with the accumulation of fat through the activation of GPR43 and GPR41, which inhibit lipolysis, thus improving the differentiation of adipose cells [330]. In this way, it is clear that an altered state of the GM not only leads to an increased susceptibility to intestinal infections but also to metabolic pathologies such as obesity, T2DM, cancer, allergies, etc. [331,332].

In recent years, various studies have focused on the possible contribution of the GM, specifically ID, to the pathogenesis of obesity. In 2006, the first study on the relationship between the intestinal microenvironment and obesity was presented by Ley et al., in which they analyzed the GM of mice with leptin deficiency. The results of this study show that the most abundant bacterial phyla were Firmicutes and Bacteroidetes, which demonstrated that obese mice with leptin deficiency carried an increased Firmicutes/Bacteroidetes ratio in comparison to the mice of normal weight [333]. Posterior studies verified that an imbalance in the proportion between Firmicutes and Bacteroidetes remains in obese human adults, in which the former decreases while the latter increases, in contrast to people of normal weight. Furthermore, an association between a loss of adipose tissue and an increase in the proportion of Firmicutes was also reported [330,334].

The traditional treatments for obesity with a nutritional focus include hypocaloric diets, bariatric surgery, physical activity, and, in some cases, pharmacotherapy. However, understanding the symbiotic relationship between the GM and metabolism in obesity could be the key to alternative treatments for this disease. Recently, diverse studies have shown the benefits of using prebiotics and probiotics in human nutrition [19,335].

Prebiotics such as fructooligosaccharides, oligosaccharides, and galactooligosaccharides have a possible beneficial role in the treatment of obesity [114,336]. The administration of prebiotics in obesity reduces lipid levels as well as the levels of inflammatory markers in serum. These effects depend on the prebiotic type and dose ingested [337].

Moreover, the use of probiotics in obesity treatment has been extensively reported. Studies conducted in experimental animal models and humans with obesity corroborate previous findings, suggesting that probiotics have a positive effect on obesity, improving weight loss, enabling the achievement of a lower BMI, and decreasing waist–hip circumference. The addition of foods rich in probiotics, such as yogurt and fermented drinks, as well as capsules and gels, boosts an increase in these bacterial populations in the intestine. The most used probiotic strains are *Bifidobacterium animalis* spp., *B*. *lactis*, *L*. *acidophilus*, *L*. *lactis* LL-23, and *L.*
*b**ulgaricus* [311,312,313,314,315,338,339].

*Lacticaseibacillus rhamnosus* GG (LGG), one of the most studied probiotic strains, is known to have benefits for glucose homeostasis. Mice treated with LGG show an improvement in insulin sensitivity and a reduction in lipid accumulation, as LGG stimulates adiponectin secretion and AMPK activation, a key enzyme in cellular energy control [316]. The use of live probiotic strains of *Lactobacillus* and *Bifidobacterium* genera on newborn mice significantly reduces body weight and visceral adipose tissue and improves insulin sensitivity [317].

The impact that prebiotics and probiotics exert on the modulation of the GM in relation to the prevention of overweight and obesity is related to a greater fermentation of SCFAs and an improvement in intestinal barrier function. Likewise, the increase in SCFAs is involved in the release of anoxygenic hormones, peptide YY, and glucagon-like peptide 1 (GLP-1) and is also related to the inhibition of triglyceride and cholesterol synthesis. The improvement of the GM helps to reduce bacterial translocation and the levels of blood LPS, thus promoting a reduction in some proinflammatory markers such as IL-6, TNF-α, and CRP [310] (Figure 5).

### 6.2. Type 2 Diabetes Mellitus

T2DM is a chronic disease that involves reduced insulin action in conjunction with a progressive loss of beta cell function [340,341]. It is a multifactorial pathology in which various pathophysiological and metabolic mechanisms are involved, which leads to a state of hyperglycemia. According to the WHO, in 2015, 1 in 11 adults suffered from T2DM worldwide, with a total of 415 million cases, of which 46.5% had not been previously diagnosed. Currently, it is estimated that by the year 2040, this number could increase to 642 million, affecting 1 out of every 10 adults [342].

There are various mechanisms involved in the development of T2DM, such as a decrease in the production and effect of incretins, insulin resistance, increased glucose absorption by the kidneys, and the altered regulation of glucose metabolism [342]. Insulin resistance is one of the factors that predisposes individuals to adipose tissue accumulation and is the result of the activation of proinflammatory mechanisms that involve the dysfunction of this tissue, which, in turn, synthesizes proinflammatory cytokines, promoting an exacerbated inflammatory phenotype [343]. In addition, other inflammatory markers are reported to be involved such as CRP, IL-6, IL-7, TNF-α, and TGF-β [344], producing a proinflammatory microenvironment.

In recent years, three mechanisms have been identified that directly influence the development of T2DM: low-grade inflammation, a decrease in amylin, and a modification of or alteration in the GM [345]. Additionally, the GM plays a vital role in the development of conditions related to diabetes mellitus, such as insulin resistance [346]. The GM also mediates immunomodulatory mechanisms through its lipid products, such as the release of LPS by Gram-negative bacteria and a decrease in SCFAs. A good intestinal barrier may prevent and control the molecular transport of dangerous metabolites through the expression of tight junction complexes; however, the highly proinflammatory microenvironment alters the expression of these proteins, leading to an increase in the activation of the inflammatory response by the GM.

The GM of patients with T2DM is very characteristic due to the reduced presence of butyrate-producing bacteria, such as *Roseburia intestinalis*. Other bacterial genera commonly associated with T2DM are *Bifidobacterium* and Bacteroidetes, and other studies have reported an association with *Faecalibacterium*, *Akkermansia*, *Roseburia*, *Ruminococcus*, *Fusobacterium*, and *Blautia* [347]. Likewise, an alteration in the bacterial proportion produces moderate dysbiosis, which promotes a proinflammatory environment through an increase in the expression of microbial genes that promote oxidative stress and that are also involved in the synthesis of vitamins, and an increase in the serum concentration of LPS and intestinal permeability [318,346].

Furthermore, as in obesity, ID associated with T2DM is related to a significantly lower presence of Firmicutes and an increase in Bacteroidetes [348]. During the course of this disease, there is an increase in opportunistic pathogens such as *Bacteroides caccae*, *Clostridium hathewayi*, *Clostridium ramosum*, and *E*. *coli* [346].

Due to this, strategies that modulate the GM have been analyzed to control or prevent T2DM, and among these, the use of probiotics has increased. The intake of probiotics not only promotes the modulation of the GM, resulting in the better fermentation of SCFAs but also improves intestinal barrier function. The increase in SCFAs leads to the release of GLP-1 and, consequently, a decrease in proinflammatory markers such as IL-6 and TNF-α, while anti-inflammatory markers are increased [318,319,349]. Studies carried out in murine models and humans have reported that the proportions of probiotic strains that belong to the phyla Firmicutes (Clostridia, Bacilli, and Negativicutes) and Bacteroidetes (Bacteroidia, Flavobacteriia, Sphingobacteria, and Cytophagia) are associated with alterations in lipid oxidation and carbohydrates [343,350,351] (Figure 5).

These results demonstrate the effect of the GM not only in the modulation of the immune response but also on the metabolism during this pathology. The use of probiotics can lead to a diminished immune response and better control of the factors related to T2DM.

## 7. Conclusions and Future Perspectives

In recent years, the GM has gained relevance as a regulator of a diversity of pathological processes, both at a local level and at distal organs. From this point onward, the so-called interorgan regulatory axes have emerged, such as the microbiota–gut–brain axis, microbiota–gut–skin axis, microbiota–gut–lung axis, and microbiota–gut–heart axis, as well as a critical relationship with metabolic diseases.

ID has been highlighted as a key element in the etiology of multiple pathologies, either directly through the production of metabolites such as SCFAs, tryptophan derivatives, and diverse neurotransmitters, or indirectly by the regulation of the immune response at an enteric system level and even at a distal organ level, for example, in the brain by allowing the passage of regulatory molecules through the BBB.

Among the main pathologies related to alterations in the regulatory axes, there are neurodegenerative diseases, developmental disorders, dermatoses, lung and heart diseases, and diseases related to metabolism. The microbiota–gut–organ alterations have created an opportunity to study different potential therapeutic agents that could treat pathologies from this approach through the regulation of the GM.

Among the therapeutic perspectives for the modulation of the GM, one of the most studied is the use of probiotic microorganisms, particularly those belonging to strains of the genera *Lactobacillus* and *Bifidobacterium*. Probiotic administration has already been proved to offer beneficial effects on different pathological processes. However, the validation of these therapies is still under research. Undoubtedly, probiotics present an important opportunity for the treatment of multiple pathologies in the near future.

Most of the proposed mechanisms that are responsible for the probiotic functions in the pathologies mentioned in this review are based on the production of microbial metabolites by the different strains that belong to the GM. By definition, these metabolites are considered postbiotics: molecules secreted by probiotic microorganisms that play a beneficial role in the health of the host when administered in adequate amounts [22]. This suggests that in the coming years, in addition to probiotic administration as a therapeutic agent in various diseases, it would be interesting to further research the use of postbiotics, which would allow for a higher degree of specificity in the treatment of pathologies. Additionally, it will also be necessary to strengthen the study of emerging approaches in the field of nanonutraceuticals, such as nanoprobiotics, as an alternative treatment that could offer greater functionality than probiotics due to its potential to generate better protection for microorganisms and ensure an increased delivery of probiotics at the intestinal level [352,353].

Finally, it is necessary to continue the study of the microbiota–gut–organs relationship to elucidate the mechanisms that favor the treatment of different pathologies and, thus, be able to identify the best probiotics to use in each of the ailments. In addition, it is still necessary to deepen the knowledge of other interorgan interactions, such as the microbiota–gut–liver axis, and to consolidate the relationship with organs, such as the kidney. Furthermore, there is a lack of research into more complex axes that involve more than one regulatory organ, such as the microbiota–gut–brain–skin axis or the microbiota–liver–brain axis [354,355].

## Figures and Tables

**Figure 1 microorganisms-10-01428-f001:**
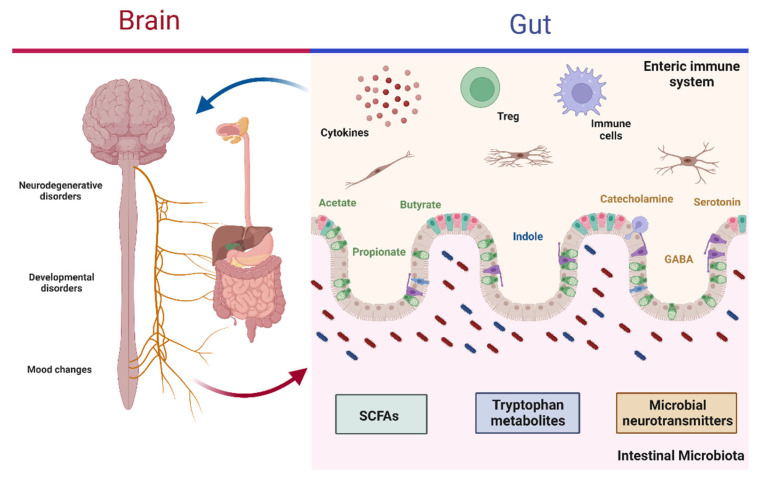
Microbiota–gut–brain axis. The interaction between the gut microbiota and CNS regulates diverse neurodegenerative diseases, developmental disorders, and mood changes. Gut bacteria produce metabolites, such as SCFAs (acetate, propionate, and butyrate), tryptophan metabolites (indole and its derivatives), and microbial neurotransmitters (GABA, serotonin, and catecholamines), which become activation signals at the ENS level in order to facilitate communication between the gut and brain via immune cells and cytokines. Depending on the composition of the gut microbiota, this may lead to protective effects when in balance or favor disease development in dysbiosis: CNS, central nervous system; SCFAs, short-chain fatty acids; GABA, γ-aminobutyric acid; ENS, enteric nervous system; Treg, regulatory T lymphocyte. Created with BioRender.com.

**Figure 2 microorganisms-10-01428-f002:**
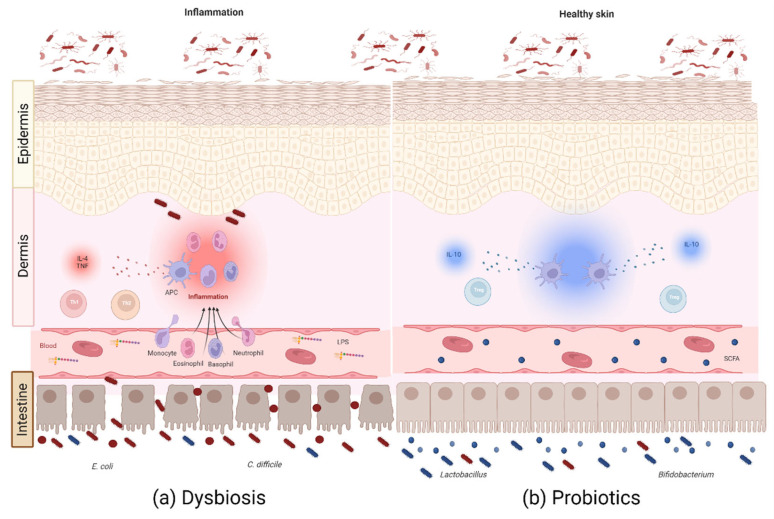
Microbiota–gut–skin axis. Bidirectional communication between the gut microbiota and skin occurs through bacterial metabolites, bacterial translocation, and immune system modulation, either by pathogens or probiotics. (**a**) Intestinal dysbiosis plays an important role in the inflammatory state of the skin in several dermatoses through the increase in proinflammatory metabolites, cytokines, and lymphocytes. (**b**) Probiotics help restore the balance through the increase in anti-inflammatory metabolites, such as SCFAs, immune cells, such as Treg lymphocytes, and cytokines, such as IL-10, to reduce and control the inflammation produced during these cutaneous pathologies: SCFAs, short-chain fatty acids; APC, antigen-presenting cell. Created with Biorender.com.

**Figure 3 microorganisms-10-01428-f003:**
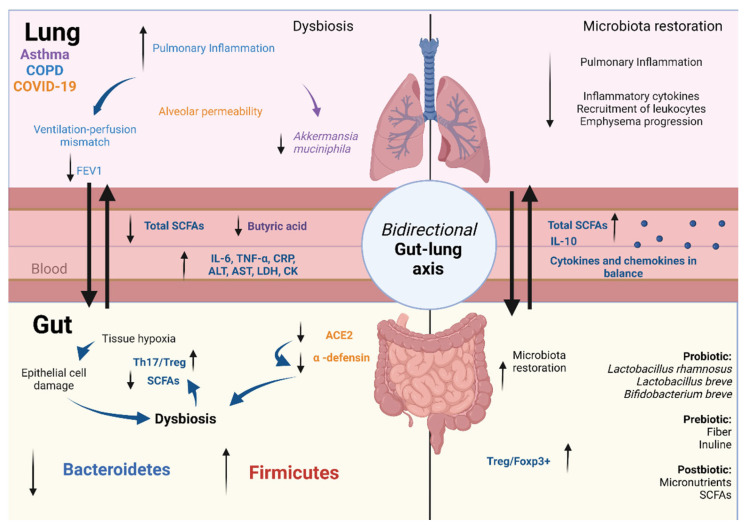
Microbiota–gut–lung axis. The microbiota–gut–lung communication is bidirectional, through the lymph and blood circulation, being conducted by metabolites released in both mucous membrane tissues. The presence of intestinal dysbiosis during lung disease plays a key role through its immunomodulatory capacity. In COPD, ventilation–perfusion mismatch leads to intestinal tissue hypoxia and epithelial cell damage associated with dysbiosis, which induces a decrease in SCFA secretion, an increase in the Th17/Treg ratio, and the secretion of proinflammatory cytokines, which correlates with pulmonary inflammation. In asthma, intestinal dysbiosis correlates positively with disease severity. The decrease in *A. muciniphila* and increase in the alveolar epithelium permeability are associated with a decrease in *Bacteroides acidifaciens* and butyric acid concentration. In COVID-19, SARS-CoV-2 decreases the ACE2 expression in the gut, which is associated with gut dysbiosis, with an increase in pathogens and a decrease in probiotic bacteria. This leads to an increase in intestinal permeability, proinflammatory markers, neutrophil recruitment, and cell activation in the pulmonary tissue. Prebiotic, probiotic, and postbiotic supplementation restores GM, reduces inflammation in the airways, restores the balance between cytokine and chemokine production, decreases leucocyte recruitment, and increases the SCFA concentration: COPD, chronic obstructive pulmonary disease; SCFAs, short-chain fatty acids; COVID-19, coronavirus disease 2019; FEV1, forced expiratory volume in the first second; ALT, alanine transaminase; AST, aspartate transaminase; LDH, lactate dehydrogenase; CK, creatine kinase; Treg, regulatory T lymphocyte. Created with Biorender.com.

**Figure 4 microorganisms-10-01428-f004:**
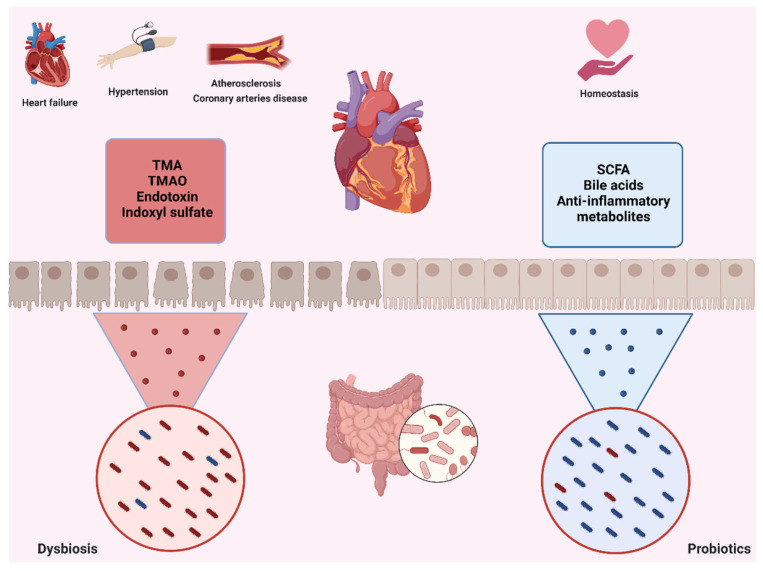
Microbiota–gut–heart axis. Through the gut microbiota and heart communication axis, ID may trigger or exacerbate heart diseases, such as hypertension, atherosclerosis, CAD, and even heart failure. In this case, ID favors the presence of bacteria that produce toxic metabolites, such as TMA, TMAO, indoxyl sulfate, and other endotoxins, which are responsible for the detrimental relationship between gut and heart. On the other hand, a balanced gut microbiota favors the production of SCFAs, bile acids, and other compounds that promote homeostatic processes in the heart: CAD, coronary artery disease; TMA, trimethylamine; TMAO, trimethylamine N-oxide; SCFAs, short-chain fatty acids. Created with Biorender.com.

**Figure 5 microorganisms-10-01428-f005:**
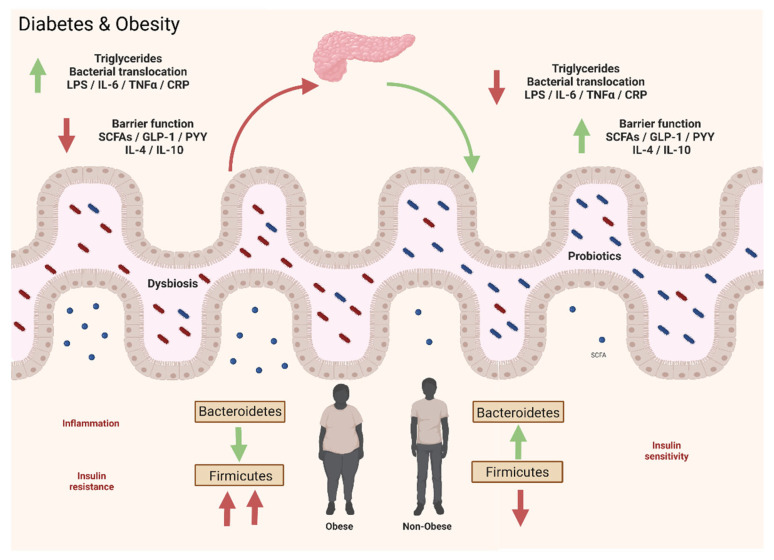
Microbiota–gut–metabolism axis. In obesity and T2DM, the inflammatory state is not produced by infections or autoimmune factors but by metabolic dysfunction in a condition known as metainflammation. The most abundant bacterial phyla of the gut microbiota are Firmicutes and Bacteroidetes; however, people with T2DM or obesity have an imbalanced proportion of these bacteria in contrast to healthy subjects. In addition, there is an increase in bacterial translocation and blood LPS, which promotes proinflammatory cytokines such as IL-6 and TNF-α. Prebiotic and probiotic intake promotes the positive modulation of the gut microbiota, produces greater saccharolytic and SCFAs fermentation, improves epithelial barrier function, and increases the abundance of anti-inflammatory markers such as IL-4 and IL-10. The increase in SCFAs impacts the release of intestinal hormones PYY and GLP-1 and is associated with the inhibition of triglyceride and cholesterol synthesis. It also reduces bacterial translocation and the production of proinflammatory markers: T2DM, type 2 diabetes mellitus; SCFAs, short-chain fatty acids; PYY, peptide YY; GLP-1, glucagon-like peptide 1; CRP, C-reactive protein. Created with Biorender.com.

**Table 1 microorganisms-10-01428-t001:** Murine models and clinical trials on the use of probiotics in ASD.

Study Model	Treatment	Effect	Ref.
Autism model in BTBR mice	*Ligilactobacillus salivarium* HA-118 and *Lacticaseibacillus rhamnosus* HA-114	Effect on positive modulation of social interaction, gut microbial diversity, and brain–gut axis signaling molecules	[76]
Autism model in C57BL/6N mice	*Bacteroides fragilis*	Improvement in intestinal permeability, alteration in gut microbiota, and improvement in communicative and sensorimotor behavior	[77]
Double-blind, placebo-controlled trial in children with ASD from Taiwan	*Lactiplantibacillus plantarum* PS128	Improvement in opposition/defiance behavior	[78]
Randomized controlled trial in children with ASD	*Lactobacillus*, *Bifidobacteria*, and *Streptococcus thermophilus*	Trial in progress	[86]
Shank3 mice model of autism	*Limosilactobacillus reuteri*	Attenuation of antisocial behavior and repetitive behaviors; modulation of GABA levels	[79]
Autistic behavior in rats	*Lactobacillus helveticus* CCFM1076 and *Lactobacillus acidophilus* JCM1132	Improvement in autistic behavior through regulation of neurotransmitter homeostasis	[80]
Model of autism in male Wistar rats	*Lactobacillus* spp. and *Bifidobacterium* spp.	Attenuation of behavioral symptoms and improvement in social behavior	[81]
Model of autism in Wistar rats	*Bifidobacterium longum* CCFM1077	Regulation of GABA neurotransmitter levels	[82]

**Table 2 microorganisms-10-01428-t002:** Murine models and clinical trials on the use of probiotics in AD.

Study Model	Treatment	Effect	Ref.
Dust-mite-induced AD in NC⁄Nga mice	*Lactiplantibacillus plantarum* CJLP55, CJLP133 and CJLP136	Decrease in AD-like skin lesions, IgE levels, eosinophil and mast cell infiltration, and IL-4 and IL-5 production; increase in IL-10 and IFN-γ and Treg cells	[153]
Ovalbumin-induced AD in SKH-1/Hr mice	*Lacticaseibacillus rhamnosus* Lcr35^®^	Decrease in AD-like skin lesions, IgE levels, inflammatory cell infiltration, and IL-4 and TSLP production; increase in Treg cells	[152]
Dust-mite-induced AD in NC⁄Nga mice	*Lacticaseibacillus rhamnosus* IDCC32 tyndallized	Improvement in AD symptoms and decrease in mast cell infiltration, IgE levels, and IL-4 production	[154]
DCNB-induced AD in NC⁄ Nga mice	*Faecalibacterium prausnitzii* EB-FPDK11 and *Akkermansia muciniphila* EB-AMDK19	Improvement in AD symptoms, skin lesions, and Th1/Th2 ratio; decrease in IgE levels, eosinophil and mast cell infiltration, and IL-4 and TSLP production; increase in filaggrin, ZO-1, and claudin-1	[155]
DNFB-induced AD in C57BL/6 mice	*Bifidobacterium longum* CCFM1029	Increased indole-3-carbaldehyde production. Inhibition of Th2 immune response; decreased TSLP, IL-4, and IL-5 production	[156]
Randomized, double-blind, placebo-controlled trial in pregnant women and infants	*Lacticaseibacillus rhamnosus* GG, *Lacticaseibacillus rhamnosus* LC705, *Bifidobacterium breve* Bb99, and *Propionibacterium freudenreichii* subsp. *shermanii* JS	Prevention of AD development	[150]
Randomized, double-blind, placebo-controlled trial in pregnant women	*Bifidobacterium bifidum* BGN4, *Bifidobacterium lactis* AD011, and *Lactobacillus acidophilus* AD031	Prevention of AD development	[151]
Randomized, double-blind trial in neonates	*Lacticaseibacillus rhamnosus* LCS-742 and *Bifidobacterium longum* subsp. *infantis* M63	Prevention of AD development	[149]
Cohort of pregnant women and infants	Probiotic milk containing *Lactobacillus acidophilus* LA-5, *Bifidobacterium lactis* Bb12, and *Lacticaseibacillus rhamnosus* GG	Reduction in AD incidence	[147]
Open trial in pregnant women and infants	*Bifidobacterium breve* M-16V and *Bifidobacterium longum* BB536	Prevention of AD development	[148]
Randomized, double-blind, placebo-controlled trial in AD patients	*Ligilactobacillus salivarius* LS01 and *Bifidobacterium breve* BR03	Improvement in AD symptoms and in Th17/Treg and Th1/Th2 ratios; reduction in microbial translocation and immune activation	[160]
Randomized, double-blind, placebo-controlled trial in children with AD	*Lacticaseibacillus rhamnosus* IDCC32 tyndallized	Improvement in AD symptoms; decrease in eosinophil cationic protein and IL-31	[162]
Randomized, double-blind, placebo-controlled trial in children with AD	*Bifidobacterium animalis* subsp. *lactis* CECT 8145, *Bifidobacterium longum* CECT 7347, and *Lacticaseibacillus casei* CECT 9104	Improvement in AD symptoms	[158]
Randomized, double-blind, placebo-controlled trial in AD patients	*Lactiplantibacillus plantarum* PBS067, *Limosilactobacillus reuteri* PBS072, and *Lacticaseibacillus rhamnosus* LRH020	Improvement in AD symptoms; decrease in TNFα, TSLP, and CCL17 levels	[159]
Randomized, double-blind, placebo-controlled trial in AD patients	*Bifidobacterium longum* CCFM1029	Improvement in AD symptoms; increased indole-3-carbaldehyde production; decreased IgE levels	[156]

**Table 3 microorganisms-10-01428-t003:** Effect of probiotics and prebiotics in mouse models and clinical trials in lung diseases.

Study Model	Treatment	Effect	Ref.
COPD model in C57BL/6 mice	*Lacticaseibacillus rhamnosus* and *Bifidobacterium breve*	Decreased inflammatory microenvironment in lung; reduction in alveolar enlargement and collagen deposition	[199]
Emphysema model in C57BL/6 mice	Cellulose and citrus pectin supplement	Beneficial modification of the intestinal microbiota and the metabolomic profile; decrease in the severity of emphysema progression	[200]
COPD model C57BL/6 and BALB/c mice	*Parabacteroides goldsteinnii* MTS01	Normalized lung function; decrease in IL-1β and TNFα expression in lung tissue and colon	[201]
Prospective cohort study in women with COPD	Total dietary fiber	Inverse association between total dietary fiber intake and the risk of COPD development	[202]
Randomized, double-blind, placebo-controlled trial in patients with COPD	Multistrain probiotic: “Vivomix 112 billion”	Improvement in muscle strength and functional performance in COPD patients by reducing intestinal permeability and stabilizing the neuromuscular junction	[203]
Randomized, double-blind, placebo-controlled trial in infants with atopic dermatitis	*Bifidobacterium breve* M-16V and a galacto/fructooligosaccharide mixture (Immunofortis^®^)	Prevention of asthma-like symptoms in infants with atopic dermatitis; decreased prevalence of frequent wheezing and noisy breathing	[204]
Ovalbumin-induced asthma model in Wistar rats	*Lactococcus lactis* NZ9000	Decrease in eosinophil infiltration, IL-4, IL-5, and IgE levels; increase in IgA, MUC-2, and claudin expression in intestine; normalization of the intestinal morphological alterations	[205]
Prenatal asthma risk model in pregnant BALB/c mice	*Bifidobacterium breve* M-16V	Decrease in eosinophil infiltration, IL-5, and IL-13 expression in neonatal mice; reduced lung inflammation in neonatal mice	[206]
Randomized, double-blind, placebo-controlled trial in infants with asthma risk	*Lacticaseibacillus rhamnosus* GG	No improvement was found	[207]
Retrospective cohort study in adults with severe COVID-19 pneumonia	Probiotic mix Sivomixx^®^, composed of: *Streptococcus thermophilus* DSM 32245, *Bifidobacterium lactis* DSM 32246, *Bifidobacterium lactis* DSM 32247, *Lactobacillus acidophilus* DSM 32241, *Lactobacillus helveticus* DSM 32242, *Lacticaseibacillus paracasei* DSM 32243, *Lactiplantibacillus plantarum* DSM 32244, and *Levilactobacillus brevis* DSM 27961	Improvement in survival rate of pneumonia	[208]
Retrospective cohort study in hospitalized adults by COVID-19	Probiotic mix Sivomixx^®^, composed of: *Streptococcus thermophilus* DSM 32245, *Bifidobacterium lactis* DSM 32246, *Bifidobacterium lactis* DSM 32247, *Lactobacillus acidophilus* DSM 32241, *Lactobacillus helveticus* DSM 32242, *Lacticaseibacillus paracasei* DSM 32243, *Lactiplantibacillus plantarum* DSM 32244, and *Levilactobacillus brevis* DSM 27961	Lower risk of respiratory failure development with resuscitation support; improvement in COVID-19-related signs and symptoms	[209]

**Table 4 microorganisms-10-01428-t004:** Murine models and clinical trials on the use of probiotics in CVD.

Study Model	Treatment	Effect	Ref.
ApoE^−/−^ mice fed with high-fat diet	*Lactobacillus acidophilus* ATCC 4356	Prevention of atherosclerosis development	[258]
ApoE^−/−^ mice fed with high-fat diet	*Bifidobacterium breve*, Bifidobacterium longum, Bifidobacterium infantis, Lactobacillus acidophilus, Lactiplantibacillus plantarum, Lacticaseibacillus paracasei, Lactobacillus bulgaricus, and *Streptococcus thermophilus*	Reduction in atherosclerotic plaques and vascular inflammation	[259]
Randomized, double-blind, placebo-controlled trial in obese postmenopausal women	*Bifidobacterium bifidum* W23, *Bifidobacterium lactis* W51, *Bifidobacterium lactis* W52, *Lactobacillus acidophilus* W37, *Levilactobacillus brevis* W63, *Lacticaseibacillus casei* W56, *Ligilactobacillus *salivarius** W24, *Lactococcus lactis* W19, and *Lactococcus lactis* W58	Decreased BP, VEGF, IL-6, TNFα, and thrombomodulin	[260]
Randomized, controlled clinical trial in subjects with metabolic syndrome	*Lacticaseibacillus casei* Shirota	No improvement found	[261]
Clinical trial in men with CAD	*Lactiplantibacillus plantarum* 299v	Improvement in vascular function; decrease in I-8, IL-12, and leptin; increase in propionate	[262]
Randomized, double-blind, placebo-controlled trial in CAD patients	*Lactiplantibacillus plantarum* 299v	Decrease in BP, leptin, IL-6, and fibrinogen levels	[263]
Rats with coronary artery occlusion	*Lacticaseibacillus rhamnosus* GR-1	Attenuation of left ventricular hypertrophy and heart failure	[264]
Randomized, double-blind, placebo-controlled trial in patients with heart failure	*Saccharomyces boulardii*	Improvement in cardiovascular function; reduction in inflammatory markers	[265]

**Table 5 microorganisms-10-01428-t005:** Murine models and clinical trials on the use of probiotics in obesity and T2DM.

Study Model	Treatment	Effect	Ref.
Randomized, double-blind, placebo-controlled trial on obese subjects	*Lactobacillus gasseri* BNR17	Decreased visceral adipose tissue (VAT) with high probiotic doses; reduction in waist circumference with both low and high probiotic doses	[311]
Randomized, double-blind, placebo-controlled trial on obese subjects	*Bifidobacterium animalis* subsp. *lactis* 420™ (B420)	Reduction in waist circumference	[312]
Randomized, double-blind, placebo-controlled trial on overweight and obese women	*Lactobacillus acidophilus*, *Lacticaseibacillus casei*, *Lactococcus lactis*, *Bifidobacterium bifidum*, and *Bifidobacterium lactis*	Reduction in the waist circumference, waist/height ratio, conicity index, and plasma PUFAs	[313]
Randomized, parallel, double-blind, placebo-controlled trial on abdominally obese subjects	*Bifidobacterium animalis* subsp. *lactis* CECT 8145 (Ba8145)	Decrease in waist circumference, waist circumference/height ratio, and BMI; increase in *Akkermansia* spp.	[314]
Randomized, parallel, double-blind, placebo-controlled trial on overweight subjects	Lab4P probiotic: *Lactobacillus acidophilus* CUL60, *Lactobacillus acidophilus* CUL21, *Lactiplantibacillus plantarum* CUL66, *Bifidobacterium bifidum* CUL20, and *Bifidobacterium animalis* subsp. *lactis* CUL34	Decrease in body weight, waist circumference, and hip circumference, but no changes in BP	[315]
High-fat-diet-induced obesity in C57BL/6 mice	*Lacticaseibacillus rhamnosus* GG	Improvement in insulin resistance; decrease in gluconeogenesis; increase in fatty acid oxidation in the liver and GLUT4 mRNA expression in skeletal muscle; enhanced adiponectin production	[316]
MSG-induced obesity in Wistar rats	Multiprobiotic Symbiter^®^ composed of 14 probiotic bacteria of genera *Bifidobacterium*, *Lacticaseibacillus*, *Lactococcus*, and *Propionibacterium*	Reduction in total body and VAT weight; improvement in insulin sensitivity; prevention of nonalcoholic fatty liver development	[317]
Randomized, double-blind, placebo-controlled trial in T2DM patients	*Lactobacillus acidophilus*, *Lacticaseibacillus casei*, *Lacticaseibacillus rhamnosus*, *Lactobacillus bulgaricus*, *Bifidobacterium breve*, *Bifidobacterium longum*, and *Streptococcus thermophilus*	Decrease in fasting plasma glucose; increase in HDL cholesterol	[318]
High-fat-diet-induced obesity in C57BL/6J mice	*Latilactobacillus sakei* OK67	Downregulation of peroxisome proliferator-activated receptor γ, fatty acid synthase, and TNFα expression in adipose tissue; decrease in hyperglycemia and inflammation; increase in tight junction proteins in colon	[319]

## Data Availability

Not applicable.

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
