# Peer review of "Probiotics: Protecting Our Health from the Gut"

_microorganisms, 2022, doi:10.3390/microorganisms10071428_

Round 1
Reviewer 1 Report
· Authors should make ‘Introduction’ with the aim of the review. Authors should emphasise the novelty of the review. What are differences between other reviews? The first paragraph is on a high level of generality – maybe Authors could rewrite it as an Introduction?
· What were the methods of the review?
· References in the text are incorrect - there should be no superscript.
· The Authors could introduce abbreviations to repeating phrases, e.g. gastrointestinal tract as GIT, etc.
· Correct font of Latin names of microorganisms/organisms to italics, e.g. lines 182; 203-204; 252-253; 256; 258; 263-264, etc. The same is with insect names, e.g. Drosophila. It should be corrected through the whole manuscript.
· Authors should apply new nomenclature for Lactobacillus species, as it was divided into 25 genera. E.g. Lactobacillus plantarum is now Lactiplantibacillus plantarum. See: LACTOTAX webpage http://lactotax.embl.de/wuyts/lactotax/ and
Zheng J, Wittouck S, Salvetti E, et al. A taxonomic note on the genus Lactobacillus: Description of 23 novel genera, emended description of the genus Lactobacillus Beijerinck 1901, and union of Lactobacillaceae and Leuconostocaceae. Int J Syst Evol Microbiol. 2020. doi: 10.1099/ijsem.0.004107.
The abbreviation for all new 25 genera is “L.”.
· Lines 185-186; 262-263; 296-298, etc. – give strain names if available. It should be corrected through the whole manuscript.
· Line 189: what species of Candida?
· Line 371 - if the Authors use the English version of clostridia, it should be written in lowercase, if in Latin, in uppercase and in italics.
· In the References list, maybe you can somehow reduce the number of authors to, for example, 10 and write further ‘et al.’, e.g. item 172, are there no such journal requirements?
Author Response
Dear Dr., we appreciate your insightful comments that help improve this article. In the following paragraphs, we will provide a point-by-point response to your comments:
- Authors should make ‘Introduction’ with the aim of the review. Authors should emphasize the novelty of the review. What are the differences between other reviews? The first paragraph is on a high level of generality – maybe Authors could rewrite it as an Introduction?
The introduction of the article was rewritten and restructured to emphasize the importance of conducting this review [Lines 32-115], and a final paragraph was added in the introduction that includes the purpose of the article. [Lines 117-123]
- What were the methods of the review?
As this is a review, not a systematic review or a meta-analysis, in our opinion a methodology section of this type is not necessary.
- References in the text are incorrect - there should be no superscript.
We have changed the reference style to use the one preferred by this journal, in which there are no superscripts used.
- The Authors could introduce abbreviations to repeating phrases, e.g. gastrointestinal tract as GIT, etc.
We have edited the manuscript to include the abbreviation GI for gastrointestinal, because it was found considerably more times than gastrointestinal tract.
- Correct font of Latin names of microorganisms/organisms to italics, e.g. lines 182; 203-204; 252-253; 256; 258; 263-264, etc. The same is with insect names, e.g. Drosophila. It should be corrected through the whole manuscript.
We have revised and corrected the manuscript to write only the correct Latin names in italics of all genera mentioned.
- Authors should apply new nomenclature for Lactobacillusspecies, as it was divided into 25 genera. E.g., Lactobacillus plantarum is now Lactiplantibacillus plantarum. See: LACTOTAX webpage http://lactotax.embl.de/wuyts/lactotax/ and,
We have corrected and rewritten using the new nomenclature for the genus Lactobacillus.
- Lines 185-186; 262-263; 296-298, etc. – give strain names if available. It should be corrected through the whole manuscript.
We have added the strains in tables for each section of the manuscript.
- Line 189: what species of Candida?
We have revised and corrected in the manuscript to add the species: Candida rugosa. [Line 209]
- Line 371 - if the Authors use the English version of clostridia, it should be written in lowercase, if in Latin, in uppercase and in italics.
We have revised and corrected in the manuscript to write the Clostridium genus. [Line 395]
- In the References list, maybe you can somehow reduce the number of authors to, for example, 10 and write further ‘et al.’, e.g. item 172, are there no such journal requirements?
|
MDPI includes in the complete reference format guide the following recommendation: Avoid including notes with the references, e.g. do not use: 10. Díaz et al. recently reported a multigram display of azide and cyanide components on a versatile scaffold, see: Díaz, D.D.; Converso, A.; Sharpless, K.B.; Finn, M.G. 2,6-Dichloro-9-thiabicyclo[3.3.1]nonane: Multigram Display of Azide and Cyanide Components on a Versatile Scaffold. Molecules 2006, 11, 212–218. Instead use the reference: 10. Díaz, D.D.; Converso, A.; Sharpless, K.B.; Finn, M.G. 2,6-Dichloro-9-thiabicyclo[3.3.1]nonane: Multigram Display of Azide and Cyanide Components on a Versatile Scaffold. Molecules 2006, 11, 212–218, doi:10.3390/11040212.
https://mdpi-res.com/data/mdpi_references_guide_v5.pdf |
Reviewer 2 Report
The novelty character of this review respect to the other present in literature should be marked.
A section Methodology reporting criteria of bibliographic research should be inserted, including a graphical scheme.
Generally the different parts of paper should be better linked among them.
Table/s with main representative studies from literature search are welcome.
The subparagraphs 2.1, 2.2, 2.3, 2.4, 2.5 should be better introduced and linked among them.
The authors should mark future directions and emerging approach such as application of nanotechnologies and mention related references such as:
Yeung et al. 2020. Big impact of nanoparticles: analysis of the most cited nanopharmaceuticals and nanonutraceuticals research. Current Research in Biotechnology, 2, November 2020, Pages 53-63.
Durazzo et al. An Updated Overview on Nanonutraceuticals: Focus on Nanoprebiotics and Nanoprobiotics. Int J Mol Sci. 2020 Mar 26;21(7):2285. doi: 10.3390/ijms21072285.
Author Response
Dear Dr., we thank you for your comments, as they have helped us to improve this manuscript. Here we answer your suggestions, point-by-point, as requested:
- The novelty character of this review respect to the other present in literature should be marked.
The introduction of the article was rewritten and restructured to emphasize the importance of conducting this review [Lines 32-115], and a final paragraph was added in the introduction that includes the purpose of the article. [Lines 117-123]
- A section Methodology reporting criteria of bibliographic research should be inserted, including a graphical scheme.
As this is a review, not a systematic review or a meta-analysis, in our opinion a methodology section of this type is not necessary.
- Generally, the different parts of paper should be better linked among them.
We have corrected and edited to show a more cohesive manuscript
- Table/s with main representative studies from literature search are welcome.
We have added new tables for the following sections of the manuscript: [Lines 448, 558, 770 and 951]
- The subparagraphs 2.1, 2.2, 2.3, 2.4, 2.5 should be better introduced and linked among them.
We have corrected and edited the manuscript to better introduce and link the aforementioned subsections.
- The authors should mark future directions and emerging approach such as application of nanotechnologies and mention related references such as:
Yeung et al. 2020. Big impact of nanoparticles: analysis of the most cited nanopharmaceuticals and nanonutraceuticals research. Current Research in Biotechnology, 2, November 2020, Pages 53-63.
Durazzo et al. An Updated Overview on Nanonutraceuticals: Focus on Nanoprebiotics and Nanoprobiotics. Int J Mol Sci. 2020 Mar 26;21(7):2285. doi: 10.3390/ijms21072285.
A section on future perspectives on therapeutic approaches such as the use of nanonutraceuticals for the management of intestinal dysbiosis has been added. [Line 1119]